



# Shipping emissions in the Iberian Peninsula and its impacts on air quality

Rafael A.O. Nunes[1], Maria C.M. Alvim-Ferraz[1], Fernando G. Martins[1], Fátima Calderay-Cayetano[2], Vanessa Durán-Grados[2], Juan Moreno-Gutiérrez[2], Jukka-Pekka Jalkanen[3], Hanna Hannuniemi[3], Sofia I.V. Sousa[1]

[1]LEPABE – Laboratory for Process Engineering, Environment, Biotechnology and Energy, Faculty of Engineering, University of Porto, Rua Dr. Roberto Frias, 4200-465, Porto, Portugal
[2]Departamento de Máquinas y Motores Térmicos, Escuela de Ingenierías Marina, Náutica y Radioelectrónica, Campus de Excelencia Internacional del Mar (CEIMAR), Universidad de Cádiz, Spain
[3]Finnish Meterological Institute, P.O. Box 503, 00101 Helsinki, Finland

Correspondence to: Sofia I.V. Sousa (sofia.sousa@fe.up.pt)

**Abstract.** Marine traffic has been identified as a relevant source of pollutants, which cause known negative effects on air quality. The Iberian Peninsula is a central point in the connection of shipping traffic between the Americas and Africa and the rest of Europe. To estimate the effects of shipping emissions inland and around the Iberian Peninsula, EMEP MSC-W model was run considering and not considering shipping emissions (obtained with STEAM3 model). Total estimated emissions of $CO$, $CO_2$, $SO_x$, $NO_x$ and particulate matter (subdivided in elementary carbon (EC), organic carbon (OC), sulphate and ash) for the study domain in 2015 were, respectively, 49 ktonnes $y^{-1}$, 30000 ktonnes $y^{-1}$, 360 ktonnes $y^{-1}$, 710 ktonnes $y^{-1}$, 4.5 ktonnes $y^{-1}$, 11 ktonnes $y^{-1}$, 32 ktonnes $y^{-1}$ and 3.3 ktonnes $y^{-1}$. Shipping emissions increased $SO_2$ and $NO_2$ concentrations especially near port areas and also increased the $O_3$, sulphate, and particulate matter ($PM_{2.5}$ and $PM_{10}$) concentrations around all over the Iberian Peninsula coastline (especially in the south coastal region). Shipping emissions were responsible for exceedances of WHO air quality guideline for $PM_{2.5}$ in areas far from the coastline, which confirms that shipping emissions can contribute negatively to air quality, both in coastal and in inland areas.

## 1 Introduction

Marine traffic has been identified as a relevant source of pollutants especially nitrogen oxides ($NO_x$), sulphur oxides ($SO_x$) and particulate matter (PM), which lead to known negative effects on air quality and health, being that its contribution is not yet well documented (Brandt et al., 2013; Corbett et al., 2007; Nunes et al., 2017b; Sofiev et al., 2018). In fact, international shipping represents around 13% and 12% of total anthropogenic emissions of $NO_x$ and $SO_x$, respectively, (IMO, 2015). Moreover, according to Klimont et al. (2017), PM emissions from international shipping contribute with about 3–4 % to global emissions, which is comparable to the contribution of road transport. As far as known, it is up to 400 km from the coast that 70% of ship emissions occur. As pollutants can be transported hundreds of kilometres towards the mainland, ships may contribute to air quality degradation in coastal areas, as well as in inland areas (Corbett et al., 2007; Eyring et al., 2009). Over



the past 10 years, interest has been growing in studying the impact on air quality of maritime emissions in cities and ports using experimental measures (Contini et al., 2011; Merico et al., 2016, 2017; Pandolfi et al., 2011; Viana et al., 2015; Wang et al., 2019) and applying air quality models (AQMs) at local, regional and global levels (Abrutytė et al., 2014; Aksoyoglu et

al., 2016; Aulinger et al., 2016; Barregard et al., 2019; Chen et al., 2017, 2018; Eyring et al., 2007; Lauer et al., 2007; Liu et al., 2017; Marelle et al., 2016; Marmer and Langmann, 2005; Matthias et al., 2016; Monteiro et al., 2018; Sotiropoulou and Tagaris, 2017). Nevertheless, the use of AQMs, such as CMAQ, WRF, CAMx, EMEP MSC-W and others entails inevitable sources of uncertainties and some limitations, mostly conditioned by the resolution of the models, the methodological limitations as a result of the complexity of air quality assessment, the quality of the meteorological data and, the reliability of

emissions inventories (Karl et al., 2019). In the last years, the activity-based method using the Automatic Identification System (AIS) has been commonly accepted as the most accurate way to estimate shipping emissions, based on the detailed information of ship specifications and the operational data. Several authors have applied this methodology, although estimations with the Ship Traffic Emission Assessment Model (STEAM) have been recognized as the best way of conducting a reliable ship emissions inventory based on ship activity (Aulinger et al., 2016; Marmer and Langmann, 2005; Nunes et al., 2017b; Russo et

al., 2018). Aulinger et al. (2016) recognized in their study that the STEAM model could be more reliable than other methods using AIS to describe ship movements. Marelle et al. (2016) evaluated emissions estimated with STEAM2 and compared them with airborne measurements from the ACCESS (Arctic Climate Change, Economy and Society) aircraft campaign. They concluded that the use of STEAM2 lead to reasonable predictions of $NO_x$, $SO_2$, and $O_3$ in comparison with ACCESS profiles. In addition, in a study performed by Nunes et al. (2017a) that reviewed studies from 2010 based on activity-based methodology

to estimate shipping emissions, STEAM model was indicated as the best procedure to predict ships power, leading to better predictions of ships movements and more reliable emission calculations. Additionally, Russo et al. (2018) reviewed and compared five different European inventories (EMEP, TNO-MACC_III, E-PRTR, EDGAR and STEAM) including or calculating emissions from shipping; this study concluded that STEAM inventory should be used for studies requiring high-resolution shipping emissions data. STEAM allows assessing emissions from each individual ship, combining highly detailed

AIS data and technical knowledge of the ships (characteristics and operative mode). STEAM is currently on its third version. From the first to the second version, carbon monoxide (CO) and particulate matter (PM) emissions were included. The method of analysing ships resistance on the water was revised and modelling of the power consumption of auxiliary engines was improved. In the third version, improvements include methods to compensate the lack of technical information of some ships and satellite data in some regions, as well as, some refinements, allowing to account legislative regulations (emission control

areas, on-board emission abatement equipment and fuel sulphur content) (Jalkanen et al., 2012; Johansson et al., 2017). The majority of the studies on the impact of shipping emissions on air quality was performed for global scales (Dalsøren et al., 2009; Eyring et al., 2007; Lauer et al., 2007), using OsloCTM2 , CMAQ and ECHAM5/MESSy1-MADE models; continental scales were also addressed (Aksoyoglu et al., 2016; Marelle et al., 2016; Sotiropoulou and Tagaris, 2017), especially the Asian region (Chen et al., 2018; Liu et al., 2017; Zhang et al., 2017), using models with coarser resolutions (CAMx, WRF/Chem,

CMAQ and GISS-E2 global models). There are only a few studies based on modelling results that considered the impacts of



shipping emissions in local scale (Abrutytė et al., 2014; Aulinger et al., 2016; Matthias et al., 2016; Monteiro et al., 2018; Vutukuru and Dabdub, 2008). Moreover, only few have used STEAM to estimate shipping emissions, namely for the North Sea (Aulinger et al., 2016; Jonson et al., 2015), Baltic Sea (Barregard et al., 2019; Jonson et al., 2015) and northern Norway region (Marelle et al., 2016). As far as known, there is only one local study that considered specifically the Iberian Peninsula

domain, evaluating the impact of maritime emissions on air quality at European and national scales using the WRF-CHIMERE modelling system for 2016 (Monteiro et al., 2018), but not using STEAM. Shipping emissions in that study were extracted from TNO-MACC_III inventory, a high spatially resolved anthropogenic emissions data source available for Europe. This inventory has a high spatially resolution data, and the MACC-III version is an updated version with a new trend analysis for emissions for international shipping, but STEAM exhibits the highest spatially resolution in their emissions and a large number

of secondary routes that do not appear in the former inventory, making emissions predicted with STEAM more precise. It also includes the disruptive changes in environmental regulations (Emission Control Areas, EU Sulphur directive) concerning sulphur in marine fuels. Moreover, it was highlighted by the MACC-III project team a clearly necessity of more research for getting data of shipping emissions (van der Gon et al., 2017; Russo et al., 2018).

The Iberian Peninsula is the most western point of the European continent and the only natural opening by sea between the

Mediterranean and the Atlantic Ocean. Considering the strategical position of the Iberian Peninsula regarding international maritime transport and the need of reducing the above referred scientific gaps, this study aimed to: i) estimate shipping emissions based on STEAM3 for 2015; ii) quantify the impacts of shipping emissions on the ambient air quality of the Iberian Peninsula using the EMEP/MSC-W model; and iii) investigate the inland regions where the European Commission air quality standards and WHO air quality guidelines were exceeded due to shipping.

## 2    Methodology and materials

### 2.1    Study area

The Iberian Peninsula is located in the southwest of Europe, mainly constituted by Portugal and Spain territories (also includes Andorra and Gibraltar). It is bordered on the southeast by the Mediterranean Sea (coastline with ≈ 1 600 km), and on the north, west, and southwest by the Atlantic Ocean (coastline with ≈ 1 650 km) being a central point in the connection of shipping

traffic between the Americas and Africa and the rest of Europe (Global Ocean Associates, 2004a, 2004b, 2004c). In fact, the Iberian Peninsula has a central position between the English Channel and the Strait of Gibraltar, which are two busiest maritime routes in the world (Columbia University Press, 2001a, 2001b). Fig: A1 shows shipping traffic lines for 2015 that shows the relevance of the Iberian Peninsula in the international shipping traffic context.

### 2.2    Shipping emissions inventory

The shipping emissions inventory for the Iberian Peninsula in 2015 was obtained from a full bottom-up approach, using STEAM. This model combines: i) the shipping activity information from the terrestrial and satellite-based Automatic





Identification System (AIS) and the technical characteristics of each ship (from HIS Markit); ii) the engine type for over ninety-thousand ships and; iii) the emission factors for each type of ship and size, engine type and mode of operation to calculate emissions from each ship. According to the above information, STEAM allows calculating the power consumptions and loads

of each engine, as well as the quantity of fuel consumed to overcome a specific speed based on the resistance of each ship (Jalkanen et al., 2009). The model also permits to calculate shipping emissions as a function of time and location (Jalkanen et al., 2012; Johansson et al., 2013, 2017). Emissions of CO, $CO_2$, $SO_x$, $NO_x$ and particulate matter (subdivided in EC, OC, sulphates and ash) were estimated for the Iberian Peninsula, from ships with an IMO number (ships for which it is mandatory using AIS equipment) and some small vessels for which the IMO number is not mandatory but with a Mobile Maritime Service

Identity (MMSI) that produced a valid response during 2015. To compare shipping emissions with land-based emissions, the sum of the annual mean emissions of $NO_x$ and $SO_x$ from the other 11SNAP sectors for the domain of this study were calculated. Shipping emissions were analysed for monthly and seasonal patterns. Seasonal patterns were based on data from: i) January, February and March called as "winter"; ii) April, May and June called as "spring"; iii) July, August and September called as "summer"; and iv) October, November and December called as "autumn".

**2.3 EMEP modelling system - configuration and evaluation**

The open-source EMEP/MSC-W chemistry transport model, version rv4.15 was used to evaluate the contribution of shipping emissions to $NO_2$, $SO_2$, $PM_{2.5}$, $PM_{10}$, sulphate and $O_3$ concentrations in the Iberian Peninsula. Model was run on a subdomain that extends from -14.25ºE to 9.05ºE and 32.15ºN to 47.35ºN, and concentrations were simulated up to approximately 400 km from the Iberian Peninsula coast. The model was designed for two scenarios: i) shipping scenario (S-SCN) considering

shipping emissions and ii) baseline scenario (B-SCN) not considering shipping emissions. Runs were made for 2015 with a horizontal resolution of 0.1°x0.1° (long-lat) and an hourly data output. Emissions (for the same year of the shipping emissions inventory) from other sources such as, industry, road traffic, public power and among other sectors, split in 11 SNAPs, were obtained from the European emission inventories that are reported under the LRTAP Convention and the NEC Directive (EMEP/CEIP, 2018). Emissions from shipping sector considered in the inventory were excluded to avoid double counting of

emissions. Moreover, it was also considered the emissions of the dust from Sahara, NOx from lightning and from forest fires from the "Fire INventory from NCAR version 1.5" (Wiedinmyer et al., 2011). The model is divided into 34 vertical layers with the lowest layer having a thickness of 50 m. Other details about the model can be found in Simpson et al. (2012) and in Norwegian Meteorological Institute (2017a). The meteorological data for 2015 were generated by the European Centre for Medium-Range Weather forecasts with the Integrated Forecast System model. According to EMEP Status Report 1/2017,

2015 was among the warmest years in Europe with temperatures reported above normal in winter and extremely high during summer in Southern Europe. Despite this, in the Iberian Peninsula temperatures below average were registered due to a persistent south-westerly flow (Norwegian Meteorological Institute, 2017b). In spring, a prolonged high pressure was established over the Iberian Peninsula leading to above-average temperatures in Portugal and Spain. In July, Spain was affected by an extraordinary and long-lasting heatwave (Norwegian Meteorological Institute, 2017b). Regarding the performance of





the model, simulations from EMEP/MSC-W are regularly evaluated against measurements in the EMEP annual reports (Norwegian Meteorological Institute, 2018). Moreover, there are several studies that compare model results with measurements and calculations with other models (Angelbratt et al., 2011; Bessagnet et al., 2016; Colette et al., 2011, 2012; Jonson et al., 2010; Karl et al., 2017; Prank et al., 2016; Soares et al., 2016) and recent studies that used the model to assess the effects of shipping emissions (Jonson et al., 2015, 2017; Turner et al., 2017).

The annual mean concentrations for each inland grid cell were compared with reference standards and guidelines (WHO and EU), aiming to evaluate exceedances and/or non-compliances of $NO_2$, $SO_2$, $PM_{2.5}$ and $PM_{10}$ due to shipping emissions. Comparisons were performed considering the international reference values for pollutants in ambient air namely: i) EU air quality standards for $NO_2$ (40 µg m$^{-3}$ for annual mean), $SO_2$ (125 µg m$^{-3}$ for daily mean), $PM_{2.5}$ (25 µg m$^{-3}$ for annual mean) and $PM_{10}$ (40 µg m$^{-3}$ for annual mean); and ii) WHO air quality guidelines for $NO_2$ (40 µg m$^{-3}$ for annual mean), $SO_2$ (20 µg

m$^{-3}$ for daily mean), $O_3$ (SOMO35 - yearly sum of the daily maximum of 8 h running average over 35 ppb in ppb per days), $PM_{2.5}$ (10 µg m$^{-3}$ for annual mean) and $PM_{10}$ (20 µg m$^{-3}$ for annual mean) (European Comission, 2018; WHO, 2018).

## 3    Results and Discussion

### 3.1  Shipping emissions – spatial and seasonal variation

Table 1 summarizes the amount of emitted air pollutants from shipping and from land-based anthropogenic sources.
Comparing $NO_x$ and $SO_x$, shipping emissions with land-based emissions, on average the first were lower than the latter. Despite this, if $NO_x$ and $SO_x$ shipping emissions were added to the land-based emissions, the total would increase by 45% and 62%, respectively. Moreover, compared with emissions from the SNAP of road transport (660 ktonnes y$^{-1}$ of NOx and 7.1 ktonnes y$^{-1}$ of SOx), the emitted amounts of $NO_x$ and $SO_x$ from shipping were 1.1 and 51.3 times higher, respectively. These results show the importance of shipping emissions for these two pollutants.

Fig: 1 shows the annual mean shipping emissions of CO, $CO_2$, $SO_x$, $NO_x$, EC, OC, sulphates and ash for the Iberian Peninsula in 2015 in a 0.1°x0.1° grid cells (approximately 10 x 10 km$^2$). As can be seen, the spatial distribution was similar for all pollutants. In general, the highest emissions were established along the west coast of the Iberian Peninsula (including all Portuguese coast), in the Strait of Gibraltar and in the Mediterranean Sea, especially close to the African coast, which is consistent with world shipping traffic density (Fig: A1). It is important to emphasise that the grid cells along the coast where
ports are located had also higher emissions due to hotelling activities. Although emissions during hotelling only represent a slight part of the total shipping emissions, port areas are significant receptors of these emissions due to the concentration of ships for long periods of time in some cases (Nunes et al., 2017a). The annual average intensities of ash, CO, $CO_2$, EC, $NO_x$, OC, sulphate and $SO_x$ emissions were 9.0E-04 tonnes/yr/km$^2$, 1.38E-02 tonnes/yr/km$^2$, 8.47 tonnes/yr/km$^2$, 1.27E-01 tonnes/yr/km$^2$, 1.97E-01 tonnes/yr/km$^2$, 3.16E-03 tonnes/yr/km$^2$, 8.04E-03 tonnes/yr/km$^2$ and 1.01E-01 tonnes/yr/km$^2$,
respectively. The annual average intensities for $NO_x$ and $SO_x$ reported from researches in Asian Region (Chen et al., 2016a, 2017; Fan et al., 2016) were in general considerably higher than those found in this study. Chen et al. (2016b) reported $NO_x$



and $SO_x$ annual average intensities of 5.06 tonnes/yr/km$^2$ and 7.14 tonnes/yr/km$^2$ respectively for Tianjin Port, located on the western shore of Bohai Bay (the largest port in northern China). Fan et al. (2016) and Chen et al. (2017) reported $NO_x$ annual average intensities of 1.0 tonnes/yr/km$^2$ and 1.83 tonnes/yr/ km$^2$ and $SO_x$ annual average intensities of 1.90 tonnes/yr/km$^2$ and

1.42 tonnes/yr/km$^2$, respectively in East China Sea and Qingdao Port (North China). It was possible to identify in the present study two main hubs given the high emissions intensity: Valencia Port and the Strait of Gibraltar. At Valencia Port, ash, CO, EC and OC had the highest values, respectively, 1.46E-01 tonnes/yr/km$^2$, 1.85 tonnes/yr/km$^2$, 1.99E-01 tonnes/yr/km$^2$ and 5.09E-01 tonnes/yr/km$^2$. At the Strait of Gibraltar, $CO_2$, $NO_x$, sulphate and $SO_x$ had the highest values, respectively, 1330 tonnes/yr/km$^2$, 24 tonnes/yr/km$^2$, 1.03 tonnes/yr/km$^2$ and 11.6 tonnes/yr/km$^2$. In accordance to what was referred above, in the

Asian Region maxima intensities were higher than those here estimated (Chen et al., 2016b; Fan et al., 2016; Ng et al., 2013). Chen et al. (2016a) reported $SO_2$ and $NO_x$ highest intensities of 1.51E+03 tonnes/yr/km$^2$ and 1.79E+03 tonnes/yr/km$^2$, respectively and Fan et al. (2016) of 1.0E+04 tonnes/yr/km$^2$ and 1.30E+03 tonnes/yr/km$^2$, respectively. Ng et al. (2013) reported $SO_2$ highest intensities of 1.1E+02 tonnes/yr/km$^2$ and 2.0E+02 tonnes/yr/km$^2$, respectively. The big differences between the average and highest emission intensities of the present study and those of the Asian studies, appear to be related

with the high intensity and type of maritime traffic and to the restrict fuel regulations in Europe. In fact, seven of the ten largest container ports in the world are located in China, and Asia is the region with the highest world seaborne trade, characterized by high traffic of container ships that have already been documented as one of the most pollutant category of ships (Chen et al., 2018; Ng et al., 2013; Nunes et al., 2017b; Song and Shon, 2014; UNCTAD, 2017). Moreover, since 2010, a 0.1% maximum sulphur requirement for fuels was stablished for ships at berth in EU ports, however, in China, there are only some

domestic emission control areas with 0.5 % maximum (Reuters, 2018).

Shipping emissions were also analysed for monthly and seasonal patterns. Seasonal amounts of shipping emissions for the pollutants analysed are shown in Table 2. According to 2015 data, the largest amounts of pollutants were emitted in summer and spring, accounting for approximately 26% of the annual total in both cases, being similar for the other seasons (23% and 25% for winter and autumn, respectively). Fig: 2 shows the monthly amounts of $CO_2$, $NO_x$ and $SO_x$ in tonne y$^{-1}$ and ash, CO,

EC, OC and sulphate in kg y$^{-1}$ of shipping emissions in the study domain during 2015. It can be observed that emissions increased progressively from February to July, where they reached the maximum annual value. After that, a decrease during August and September was observed, followed by a stabilization during October (for some pollutants there was a slight increase) and a decrease until December. Although emissions varied throughout the year, variations were about 1-2% between months and each month represented 7.1-9.1% of the annual total emissions. These slight variations seem to be related to the

navigation conditions (better during the spring and summer), which consequently increases the number of ships that sails in this zone (during May, June, July and August). Seasonal and monthly patterns here studied are consistent with results reported by Corbett et al. (1999), which described that $NO_x$ and $SO_2$ global shipping emissions had slight variations during the year with each month representing 7 to 9% of the annual total emissions, similarly to what was estimated in the present study. This result contradicts what was expected, because as far as is known each region has different trading seasons with a strong seasonal

influence mainly due to the influence of the weather in the navigability conditions. According to this result, the Iberian





Peninsula seems to be located where globally variable shipping lanes share common routes, resulting in a regional pattern similar to the global average pattern. Fan et al. (2016) also reported slight seasonal variations similar to this study, although for East China Sea higher emissions were verified during spring. Jalkanen et al. (2009) reported higher shipping emissions during summer (highest emissions during July) for Baltic Sea in 2007 and a similar seasonal variation pattern, although the

variation was higher (20% between the months with the highest and lowest $NO_x$ emissions).

### 3.2 Impact on Air Quality

To understand shipping emissions impact on air quality over the Iberian Peninsula in 2015, EMEP model was configured considering and not considering shipping emissions. Fig: 3 shows the contribution of shipping emissions to the annual average of $NO_2$, $SO_2$, sulphate, $O_3$, $PM_{2.5}$ and $PM_{10}$ concentrations in the Iberian Peninsula. Results show that when shipping emissions

were considered, the concentrations of $NO_2$, $SO_2$, sulphate, $PM_{2.5}$ and $PM_{10}$ increased, especially in the Strait of Gibraltar and close to the coastal areas (mainly in port areas) as well as along the west coast of the Iberian Peninsula, (along main shipping routes). $O_3$ concentrations also increased due to shipping emissions especially in the Mediterranean Sea close to the African coast. An opposite behaviour was verified with a decrease of concentrations around the major shipping routes in the west coast of the Iberian Peninsula, close to the southern coastal area of Spain and in some port areas as a result of $NO_x$ titration caused

by increased $NO_x$ shipping emissions. Aksoyoglu et al. (2016) also reported an increase in the mean $O_3$ concentrations of 5–10 % in the Mediterranean Sea and a decrease of the levels around some major ship lanes (English Channel and North Sea). Moreover, Merico et al. (2016) that performed experimental measurements in a port-city in Italy found correspondence between NO peaks and $O_3$ titration. Thus shipping emissions have the potential to decrease $O_3$ concentrations close to the main ship lanes and ports and increase at larger distances from the emissions source, which seems to be a local scale effect. Annual

mean concentrations when shipping emissions were included (considering all grid cells of the domain) of $NO_2$, $SO_2$, sulphate, $O_3$, $PM_{2.5}$ and $PM_{10}$ were, respectively, 1.8 µg m$^{-3}$, 0.5 µg m$^{-3}$, 0.8 µg m$^{-3}$ (mean increase of 67%), 80 µg m$^{-3}$, 8.2 µg m$^{-3}$, 22 µg m$^{-3}$. The highest differences in the annual mean concentrations of $NO_2$, $SO_2$, sulphate, $O_3$, $PM_{2.5}$ and $PM_{10}$ w/ship case and wt/ship case were 31.7 µg m$^{-3}$, 16.1 µg m$^{-3}$, 3.4 µg m$^{-3}$, 13 µg m$^{-3}$, 4.8 µg m$^{-3}$ and 6.9 µg m$^{-3}$, respectively. Monteiro et al. (2018) reported for Europe similar differences of $PM_{10}$ (7 µg m$^{-3}$) and of $O_3$ (14 µg m$^{-3}$) and lower of $NO_2$ (18 µg m$^{-3}$) evidenced

in the main shipping routes of Straits of La Mancha and Gibraltar. The higher $NO_2$ concentrations reported in this study compared with Monteiro et al. (2018) seem to be related to the shipping emissions inventory that was used. As was already mentioned the shipping emissions used by Monteiro et al. (2018) were extracted from TNO-MACC_III inventory which seems to underestimate $NO_x$ emissions of this sector compared with the STEAM (more precise). Aksoyoglu et al. (2016) reported lower differences for $PM_{2.5}$ (3.5 µg m$^{-3}$) and $O_3$ (12 µg m$^{-3}$) for Europe. Fig: 4 shows the relative impact of shipping emissions

on pollutant concentrations for the Iberian Peninsula. Locally, the effects were more evident in the sea areas along main shipping routes and especially in the Strait of Gibraltar and in the Mediterranean Sea, with contributions of more than 90% for $NO_2$ and $SO_2$, 80% for sulphate, 25-50% for $PM_{2.5}$ and 20-35% for $PM_{10}$. Regarding $O_3$, shipping emissions contributed to an increase of around 15% all over the Iberian Peninsula coastline and in the Mediterranean Sea close to the African Coastline.





Nevertheless, shipping emissions also contributed to decrease $O_3$ concentrations around 15-40% in the Strait of Gibraltar and
close to Valencia Port. It is also important to emphasise that along main shipping routes and close to the Iberian Peninsula port
areas (except Valencia), $O_3$ concentrations considering and not considering shipping emissions remained the same. Aksoyoglu
et al. (2016) found higher contributions (45%) for $PM_{2.5}$ and lower increases contributions (5-10%) for $O_3$ in the Mediterranean
Sea. Sotiropoulou and Tagaris (2017) also reported contributions higher than 90% for $NO_2$ and $SO_2$ and 40% during winter
and 50% during summer for $PM_{2.5}$ over the Mediterranean Sea. Viana et al. (2014) reviewed studies concerning the impact of
shipping emissions on air quality in European coastal areas and reported lower contributions than those estimated in this study
for the Strait of Gibraltar (2-4% for mean annual $PM_{10}$ and 14% for mean annual $PM_{2.5}$) and Southern Spain close to Bay of
Algeciras (3-7% for mean annual $PM_{10}$ and 5-10% for mean annual $PM_{2.5}$). The differences between the contributions reported
by Viana et al. (2014) seem to be related to the methodology used in the reviewed studies (source apportionment of $PM_{10}$ and
$PM_{2.5}$ by positive matrix factorization). Although the impact of shipping emissions on pollutants' concentrations has been most
evident in sea areas, they also contributed to increasing inland concentrations. As shown in Fig: 4, shipping emissions generally
contributed to about 50% of inland $NO_2$ concentrations near port areas of Portugal and Spain, reaching more than 75% in the
province of Cadiz. Similar behaviour was observed for $SO_2$ concentrations, however, in this case, contributions of more than
75% were also noticed in the province of Malaga. As already mentioned, for $O_3$, contributions of around 5-15% were calculated
for the entire Iberian Peninsula coastline, especially in the south coastal region. Regarding sulphate, contributions of around
60% were calculated for all the Iberian Peninsula south coastal region, with contributions of 20-40% when all the Iberian
Peninsula was considered. For $PM_{2.5}$ and $PM_{10}$, the highest contributions (around 20-30%) were also verified in the Iberian
Peninsula south coastal region. When all the Iberian Peninsula was considered, $PM_{2.5}$ and $PM_{10}$ contributions were 10% and
15%, respectively. Monteiro et al. (2018) reported for the west coast of Portugal (also the west coast of Iberian Peninsula)
lower contributions for $NO_2$ and $PM_{10}$ (higher than 20% and less than 5%, respectively) than those reported in this study
probably due to the different methodology applied.

The higher contributions of shipping emissions for pollutant concentrations in coastal regions (mainly to $NO_2$ and $SO_2$
concentrations) as well as in inland regions (sulphate, $O_3$, $PM_{2.5}$ and $PM_{10}$ concentrations) indicates that ships are a non-
negligible source. According to the model results, in general, the higher contributions of shipping emissions to the
concentrations levels were registered during summer and spring periods. This pattern seems to be related to the increase of
ship traffic during summer due to better navigation conditions at this time of year, which increases the emissions and
subsequently atmospheric pollution. Moreover, during summer months, the number of passenger ships tends to increase (due
to recreational travel), especially in the Mediterranean Sea, which may have led to an increase of shipping emissions and their
contributions to the pollutant's concentration levels. Results were consistent with those achieved by Aksoyoglu et al. (2016),
Chen et al. (2017), Sotiropoulou and Tagaris (2017) and Chen et al. (2018), which also reported largest contributions of
shipping emissions on $PM_{2.5}$, $O_3$, $NO_2$ and $SO_2$ concentrations during summer.

Fig: 5 shows $NO_2$, $SO_2$, $O_3$, $PM_{2.5}$ and $PM_{10}$ exceedances to EU air quality standards and WHO air quality guidelines in the
inland regions due to shipping, as well as the differences between SOMO35 levels (in ppb.days) considering and not



considering shipping emissions. Results showed no exceedances to EU annual limit standard for $SO_2$, $PM_{2.5}$ and $PM_{10}$. Regarding $NO_2$, as the annual limit for the EU air quality standards and the WHO air quality guidelines are the same (40 µg

$m^{-3}$), the analyses were joined. As can be seen from Fig: 5 a), exceedances due to shipping emissions were verified in Valencia close to Valencia Port and in Barcelona close to Port of Barcelona. When shipping emissions were considered $PM_{2.5}$ WHO air quality guideline (10 µg $m^{-3}$) exceedances increased 7%. As can be seen from Fig: 5 b), exceedances were verified in Portugal (close to the Ports of Viana do Castelo, Leixões, Lisboa and Setúbal), across all Spanish coastline, in the north (in Pontevedra Province close to Port of Vigo and in Asturias Province close to the ports of Aviles and Gijon) and in the south (in the regions

of Andalusia close to Algeciras, Malaga and Adra Ports, in Valencia close to Port of Valencia and in the Catalonia close to Port of Barcelona) where the contribution of shipping emissions to the increase of concentrations was even more pronounced. It should be noted that shipping emissions were still responsible for exceedances in areas far from the coastline, as was verified in Viana do Castelo and more pronounced in the region of Andalusia. These results confirm that shipping emissions can contribute negatively to air quality, both in coastal and in inland areas. $PM_{10}$ WHO air quality guideline of 20 µg $m^{-3}$ was

exceeded 8% more when shipping emissions were considered. As can be seen from Fig: 5 c), exceedances were verified mainly across the southern Spanish coastline, in the regions of Andalusia and Catalonia. The contribution of shipping emissions to the increment of number of exceedances (in terms of concentrations $\Delta$ µg$m^{-3}$) of $NO_2$, $PM_{2.5}$ and $PM_{10}$ was also determined. This information can be found in Fig: A2 a), b) and c), respectively. Regarding WHO air quality guideline for $SO_2$, as the value refers to the average daily concentrations, the results are presented as the number of days per year that the threshold value was

exceeded in a given grid cell when shipping emissions were considered but were not exceeded without shipping emissions. Fig: 5 d), shows that exceedances were verified in Setúbal District (Portugal) close to Port of Sines, across all Spanish coastline, in the north (in Coruña Province close to Port of Coruña, in Asturias Province close to the ports of Aviles and Gijon and close to Port of Bilbao) and in the south (in Huelva close to Port of Huelva, in Cadiz Province close to the Strait of Gibraltar, in Valencia close to Port of Valencia, in Castellón close to Port of El Grao, in Tarragona close to Port of Tarragona and in

Barcelona close to Port of Barcelona). In the Strait of Gibraltar, it was calculated the highest number of days per year where WHO reference value for $SO_2$ was exceeded (maximum increment of 96 days). The spatial distribution of the number of days per year in which the WHO reference value for $SO_2$ was exceeded can be found in Fig: A2 d). According to the above results, mitigations measures should be studied and implemented to reduce shipping emissions mainly close to port areas, in the south of the Iberia Peninsula close to the Strait of Gibraltar and in the Mediterranean Sea. Implementing an ECA in the Mediterranean

Sea can contribute to reduce shipping emissions and help these regions to attain WHO and EU standards. As SOMO35 is an indicator of health impact assessment recommended by WHO, differences between the levels considering and not considering shipping emissions were calculated to evaluate the contribution of these emissions for the $O_3$ inland concentrations. As it can be seen from Fig: 5 e), SOMO35 levels were negative close to the Portuguese ports of Lisboa and Setúbal and close to Spanish ports of Algeciras (Strait of Gibraltar), Valencia and Barcelona. The major contributions were calculated for the southwest

coastline of the Iberian Peninsula, with levels from 500-1000 ppb.days up to 200 km from the coastline (over all south region





of Portugal), which might be explained by the highest solar radiation intensity that is felt in the southern regions of the Iberian Peninsula.

### 3.3  Uncertainties and Limitations

Given the complexity of any chemical transport model, it is difficult to specify the source of uncertainties, these are inherent
to the uncertainties of the meteorological data, emission inventory and the imperfections of chemical mechanism and physical process on the modelling system. Nevertheless, it is known that the reliability of the emissions inventory is a major cause of uncertainty. Efforts to reduce uncertainties were made by using shipping emissions input data as accurate as possible, estimated by an improved version of STEAM model (STEAM3), that has the highest spatially detailed shipping emissions inventory and have been recognized as one of the best to estimate emissions from maritime traffic (Nunes et al., 2017b; Russo et al., 2018).
Keeping the uncertainties of the atmospheric dispersion simulations in mind, efforts were made to run the EMEP-MSC/W model as accurate and detailed as possible (horizontal resolution of 0.1°x0.1°, 34 vertical levels and data output time steps of 1 h). Moreover, EMEP-MSC/W model has been recently compared with the CMAQ and the SILAM models and showed the best spatial correlation of annual mean concentrations for $NO_2$, $SO_2$ and $PM_{2.5}$ resulting of shipping emissions, although it seems to be underestimating $PM_{2.5}$ concentrations and overestimating $O_3$ concentrations. The EMEP-MSC/W model considers
the $O_3$ loss by $NO_x$ titration, the sunlight effects and $NO_x$ to VOC ratio that promotes $O_3$ production, which is an approximation allowing to minimize the effects of the non-linear $O_3$ chemistry. Moreover, estimations were performed using meteorological data from the European Centre for Medium-Range Weather Forecasts (ECMWF) for 2015.

### 4    Conclusions

In this study, Ship Traffic Emission Assessment Model (STEAM3) was used to estimate shipping emissions in the Iberian
Peninsula Region in 2015. According to the results, total estimated emissions for CO, $CO_2$, $SO_x$, $NO_x$ and particulate matter (subdivided in elementary carbon (EC), organic carbon (OC), sulphates and ash) were 49, 30000, 360, 710, 4.5, 11, 32 and 3.3 ktonnes $y^{-1}$, respectively. The highest emissions were estimated along the west coast of the Iberian Peninsula, in the Strait of Gibraltar and in the Mediterranean Sea. The largest amount of emissions for all pollutants were emitted during summer and spring (reaching the maximum during July) which seemed to be related to the navigation conditions. The estimated shipping
emissions were equivalent to 45% and 62% of $NO_x$ and $SO_x$ of the total land-based emissions, respectively, which shows that shipping emissions cannot be neglected. Running the EMEP/MSC-W model it was possible to verify that the effects of shipping emissions on air quality were more evident in the sea areas along the main shipping routes and especially in the Strait of Gibraltar and in Mediterranean Sea. Although the contribution of shipping emissions to pollutants concentrations has been more evident in sea areas, they also contributed to increasing the inland concentrations. It was verified that shipping emissions
increased $SO_2$ and $NO_2$ concentrations around 50% near port areas of Portugal and Spain, reaching more than 75% in the provinces of Cadiz and Malaga, $O_3$ concentrations around 5-15% for all the Iberian Peninsula coastline, especially in the south



coastal region and sulphate, and particulate matter ($PM_{2.5}$ and $PM_{10}$) concentrations around 60% and 20-30%, respectively all over the Iberian Peninsula south coastal region. $NO_2$, exceedances due to ship emissions were verified in Valencia and Barcelona. WHO air quality guideline for $PM_{2.5}$ and $PM_{10}$ were exceeded, respectively, 7% and 8% more when shipping

emissions were considered. In the regions close to the Strait of Gibraltar were verified the highest exceedances of WHO air quality guideline for $SO_2$ (maximum increment of 96 days). The major contributions of shipping emissions to inland SOMO35 levels were for the southwest coastline of the Iberian Peninsula, with levels of 500-1000 ppb.days up to 200 km from the coastline (overall south region of Portugal). These results confirm that shipping emissions can contribute negatively to air quality, both in coastal and in inland areas and mitigations measures should be studied and implemented to reduce shipping

emissions mainly close the port areas and in the south of the Iberia Peninsula (close to the Strait of Gibraltar and in the Mediterranean Sea). In the future, it is important to study the impacts of shipping emissions on health, which are still underestimated and rarely studied.

## Code availability

The EMEP model is available as Open Source (see https://github.com/metno/emep-ctm code version rv4.17 (201802) (EMEP

MSC-W, 2018, https://doi.org/10.5281/zenodo.3355023).

### Author contributions

RAON performed EMEP MSC-W simulations and the analysis, did the interpretation of the results and wrote the manuscript. JPJ and HH provided the ship emission data. MCMA, FGM, JMG and JPJ reviewed the paper and helped in the interpretation of the results. FCC, VDG, and JMG gave support in the interpretation of the results for Spain and reviewed the paper. SIVS

designed the study and assisted in modelling scenarios and in writing the paper.

### Competing interests

The authors declare that they have no conflict of interest.

### Acknowledgements

This work was financially supported by: project UID/EQU/00511/2019 - Laboratory for Process Engineering, Environment,

Biotechnology and Energy – LEPABE funded by national funds through FCT/MCTES (PIDDAC); project "LEPABE-2-ECO-INNOVATION" – NORTE-01-0145-FEDER-000005, funded by Norte Portugal Regional Operational Programme (NORTE 2020), under PORTUGAL 2020 Partnership Agreement, through the European Regional Development Fund (ERDF) and project EMISSHIP (POCI – 01 – 0145 – FEDER – 032201), funded by FEDER funds through COMPETE2020 – Programa



Operacional Competitividade e Internacionalização (POCI) and by national funds (PIDDAC) through FCT/MCTES. Dr
Jalkanen would like to acknowledge the financial support from the Nordic Council of Ministers project EPITOME (Emissions
from shiP and the Impacts on human healTh and envirOnMEnt - now and in the future), project number KOL-1601.

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





**Table 1.** Annual mean amounts of emitted air pollutants from shipping and from land-based anthropogenic sources during 2015 (in tonne $y^{-1}$)

| Pollutant | Shipping | Land-based emissions [a] | Road transport emissions |
|:---:|:---:|:---:|:---:|
| Ash | 3.3E+03 | - | - |
| EC | 4.5E+03 | - | - |
| OC | 1.1E+04 | - | - |
| $NO_x$ | 7.1E+05 | 1.6E+06 | 6.6E+05 |
| $SO_x$ | 3.6E+05 | 5.8E+05 | 7.1E+03 |
| Sulphate | 3.2E+04 | - | - |
| $CO_2$ | 3.0E+07 | - | - |
| CO | 4.9E+04 | 3.6E+06 | 5.7E+05 |
| Total | 3.1E+07 | - | - |

[a] Emissions from 11 SNAP sectors, namely, public electricity and heat production, industry, other stationary combustion sources, fugitive emissions, solvents, road transport, aviation, off-road sources, waste, agriculture livestock, agriculture other sources and other sources.



**Table 2.** Seasonal amounts of emitted air pollutants from shipping in the Iberian Peninsula in 2015 (in tonne $y^{-1}$)

| Pollutant | Spring | Summer | Autumn | Winter | Total |
|:---:|:---:|:---:|:---:|:---:|:---:|
| **Ash** | 0.85 | 0.87 | 0.83 | 0.77 | 3.3 |
| **EC** | 1.2 | 1.2 | 1.1 | 1.0 | 4.5 |
| **OC** | 2.9 | 3.0 | 2.8 | 2.6 | 11 |
| **NO$_x$** | 1.8E+02 | 1.9E+02 | 1.8E+02 | 1.6E+02 | 7.1E+02 |
| **SOx** | 92 | 94 | 91 | 85 | 36 |
| **Sulphate** | 8.3 | 8.4 | 8.1 | 7.6 | 32 |
| **CO$_2$** | 7.8E+03 | 8.0E+03 | 7.6E+03 | 7.0E+03 | 3.0E+04 |
| **CO** | 13 | 13 | 12 | 12 | 49 |
| **Total** | 8.3E+03 | 8.3E+03 | 7.9E+03 | 7.2E+03 | 3.1E+04 |



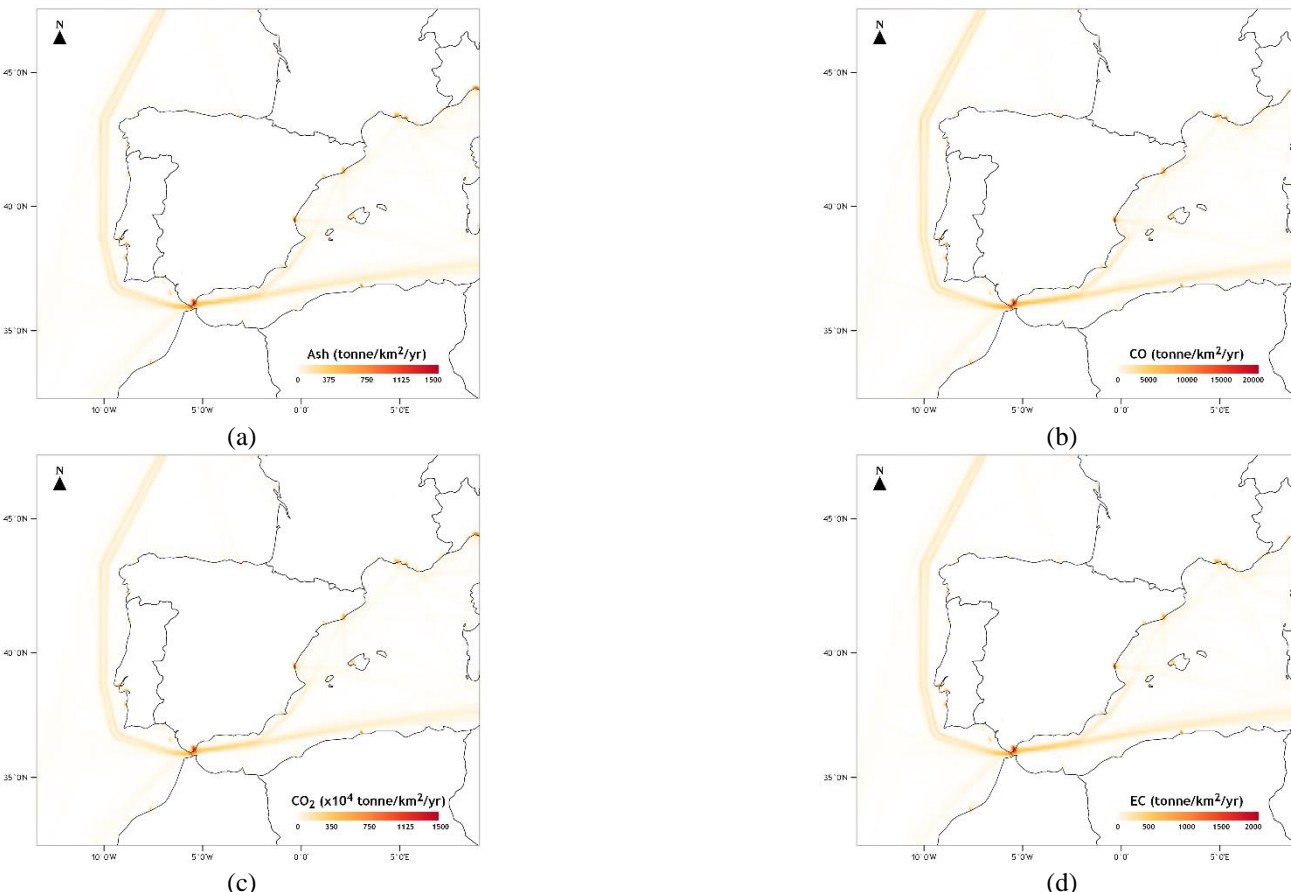

**Figure 1:** Shipping emissions of a) ash; b) CO; c) $CO_2$; d) EC; e) $NO_x$; f) OC; g) sulphate and h) $SO_x$ in the study domain for 2015






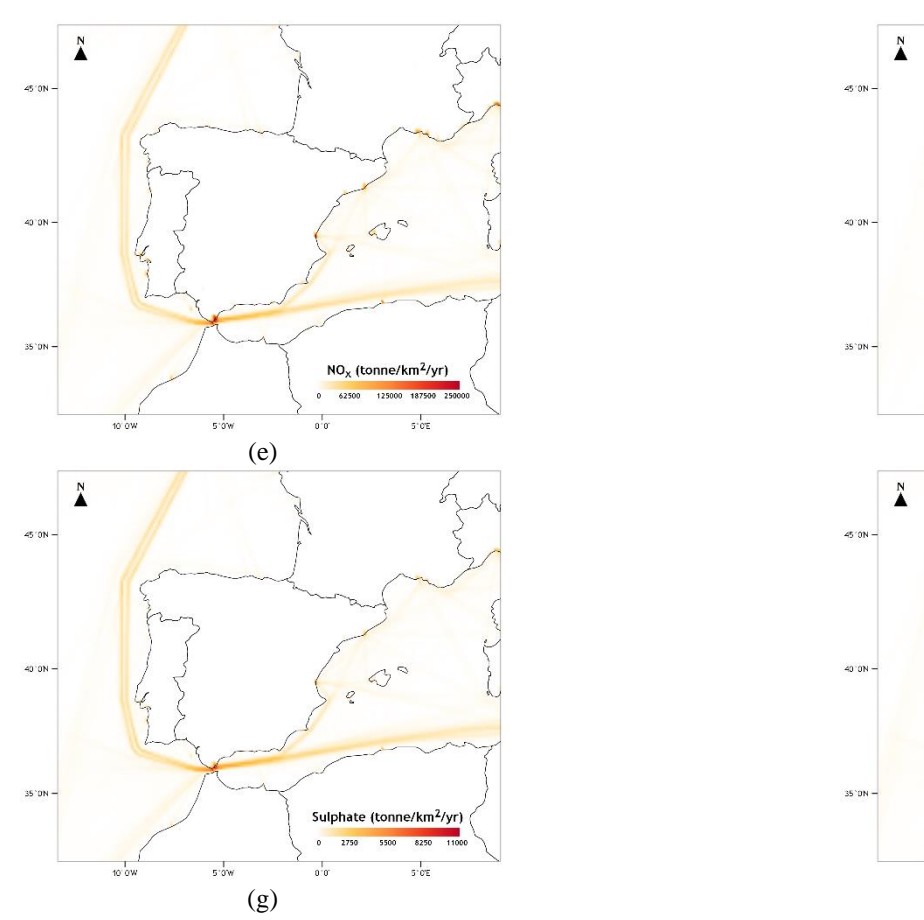

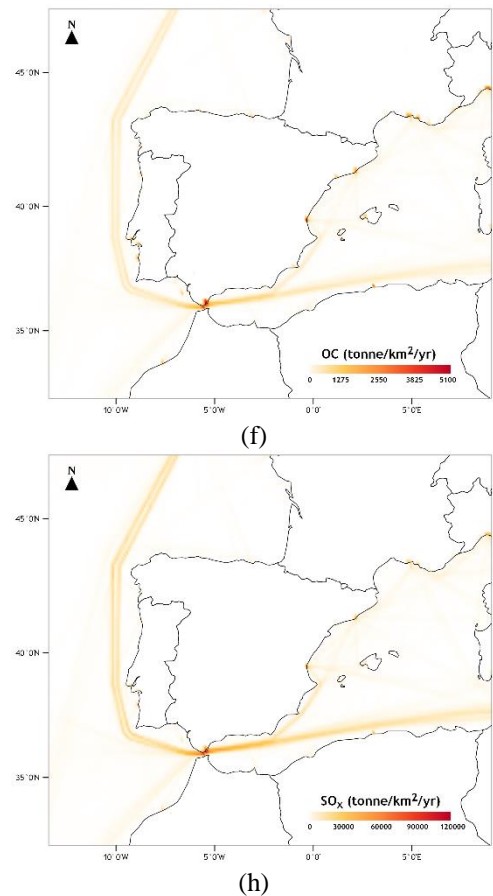

**Figure 1 (continued):** Shipping emissions of a) ash; b) CO; c) $CO_2$; d) EC; e) $NO_x$; f) OC; g) sulphate and h) $SO_x$ in the study domain for 2015





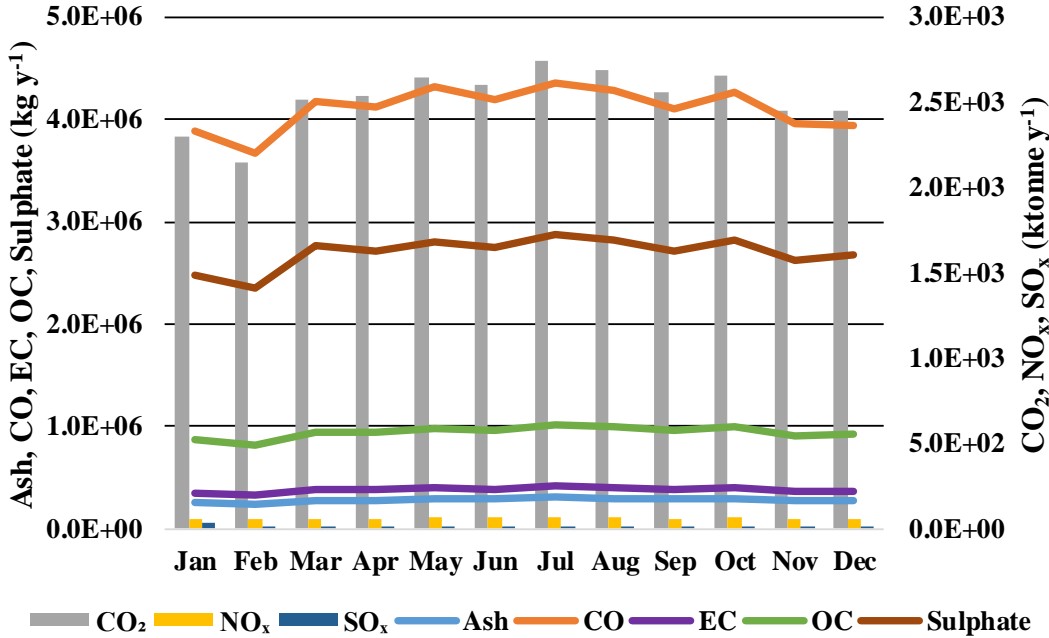

**Figure 2:** Monthly amounts of $CO_2$, $NO_x$ and $SO_x$ in ktonne $y^{-1}$ (bars-right axis) and ash, CO, EC, OC and sulphate in kg $y^{-1}$ (lines-left axis) shipping emissions in the study domain during 2015





**Figure 3:** Contribution of shipping emissions to annual mean concentrations of a) NO₂; b) SO₂; c) PM₂.₅; d) PM₁₀; e) sulphate; f) O₃ in the study domain in 2015 (Δ=S-SCN - B-SCN)





**Figure 4.** Contribution of shipping emissions (%) to annual mean concentrations of a) $NO_2$; b) $SO_2$; c) $PM_{2.5}$; d) $PM_{10}$; e) sulphate and f) $O_3$ for 2015

**Figure 5:** Spatial distribution of the inland exceedances for a) $NO_2$ to EU air quality standards and WHO air quality guidelines (same value); b) $PM_{2.5}$; c) $PM_{10}$ and d) $SO_2$ to WHO air quality guidelines and e) SOMO35 levels due to shipping emissions contributions.





**Appendix A**

This appendix contains the spatial distribution of world shipping traffic density and a zoom of the study area for 2015, as well as the spatial distribution of the contribution of shipping emissions in terms of concentration increment to the inland exceedances.





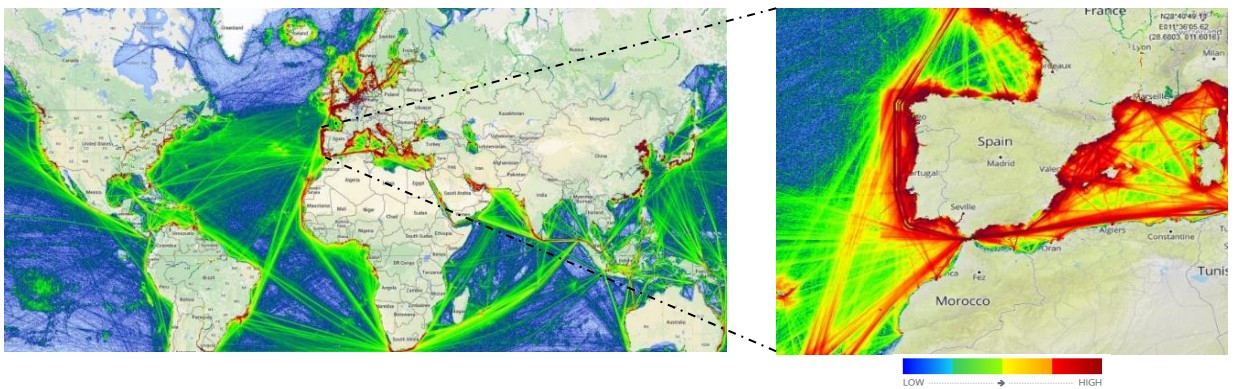

**Figure A1.** World shipping traffic density map and a zoom of the study area for 2015 (source: Marine Traffic, 2016).





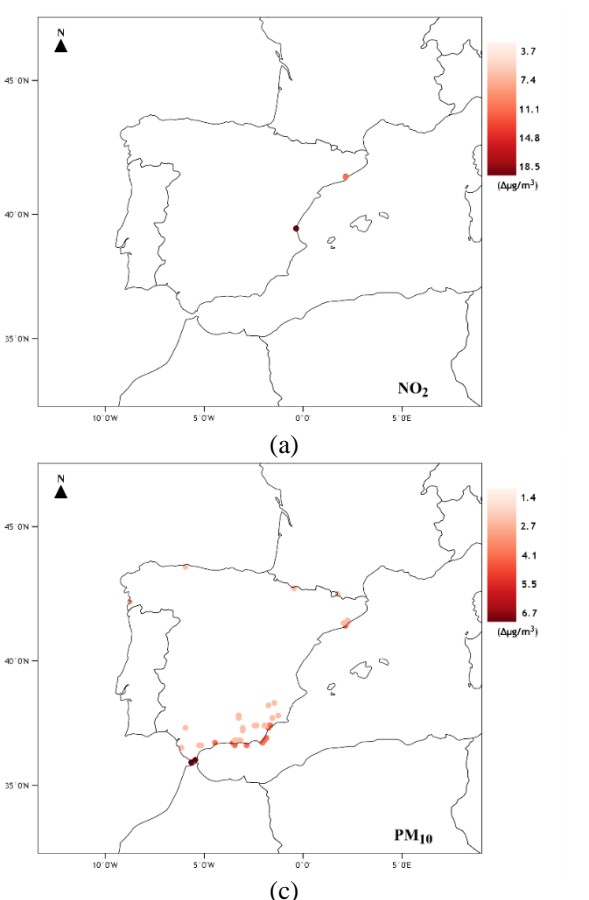

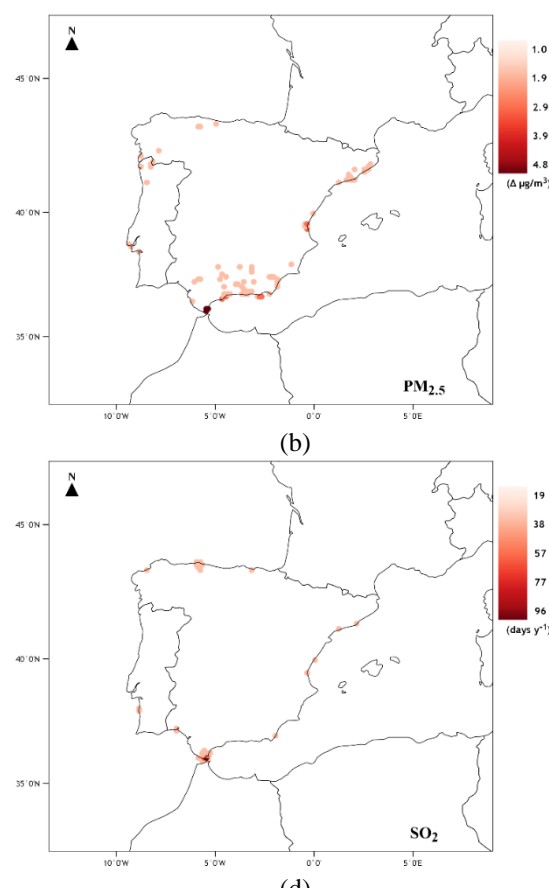

**Figure A2. –**Spatial distribution of the contribution of shipping emissions (concentration increment) to the inland exceedances for a) NO$_2$; b) PM$_{2.5}$ and c) PM$_{10}$ and number of days per year that the threshold value was exceeded in a given grid cell for d) SO$_2$.