# Peer review of "Shipping emissions in the Iberian Peninsula and its impacts on air quality"

_Atmospheric Chemistry and Physics, 2019_

## Referee Comment (RC1) · Anonymous Referee #1 · 26 Feb 2020

Dear Editor,

this MS models shipping emissions across the Iberian Peninsula andtheir impacts on air quality. The MS is mostly well written and straightforward, and interesting for the scientific community even if certain statements (e.g., the last sentence in the abstract "confirms that shipping emissions can contribute negatively to air quality") are already widely established. I have two main concerns:

1) Model validation: no evidence is preported that the model was validated with observations. Was it validated in any way? How? Even if it cannot be validated with actual shipping contributions due to the different methods used (lines 237-238), the authors could compare their results to total $NO_2$ or other gaseous pollutant concentrations from reference stations, for example. This kind of comparison would be essential to confirm

their modelling results.

2) Primary vs. secondary aerosol contributions: it is unclear in the manuscript whether the particle concentrations modelled are primary or primary + secondary aerosols from shipping. If secondary aerosols were included, how was this implemented in the model? This is the main limitation of most models targeting shipping emissions (both dispersion and receptor models). Please address this carefully in the Methods section.

Specific comments: - line 25, "its contribution", does this refer to health impacts? The contribution to air quality degradation has been assessed in numerous papers in the literature, including the papers referenced by the authors.

- the English could be reviewed by a native speaker, it is good but some small typos remain.

- line 33, suggestion to reference the EEA report EEA, 2013. The impact of international shipping on European air quality and climate forcing. EEA Technical Report 04/2013. Luxembourg: Publications Office of the European Union, 2013. ISBN 978-92-9213-357-3.

- Please add in the Methods section discussions on model validation and on secondary aerosols (whether they are or not included in the model).

- line 150, what does "ash" refer to, exactly? Please define

- line 155, "ports", the resolution is quite coarse (10x10 km2) to represent harbour emissions, or even most coastal urban areas. Please highlight this as a limitation

- line 160, suggest to check and reference the report HEI Special report 22, Impacts of shipping on air pollution emissions, air quality, and health in the Yangtze River Delta and Shanghai, China

- lines 185-188, please add a statistical trend analysis: the differences don't seem

statistically significant, to the naked eye.

- line 191: I don't think the comparison with a paper from 1999 (even if a reference paper) is adequate here: in 20 years the trade and sailing patterns have surely changed largely, therefore this comparison is not representative

- line 203, are these primary or secondary PM10 and PM2.5 concentrations? Or the sum of both?

- line 206, why do concentrations increase gradually towards and over the N of Africa? Are there no O3 sinks (e.g., major cities) in this region? This seems unlikely, probably the emission inventories are not accurate for this region. Please discuss.

- line 217, are these (4.8 microg/me and 6.9 microg/m3) shipping contributions?They seem quite high, especially if only primary aerosols are considered (I'm still unsure of this). Also, are these average values for the entire peninsula? Please compare with shipping contributions from the literature, and also with total (non-ship sourced) PM10, NO2, O3, etc concentrations.

- line 240-241: these contributions seem unlikely as they are reported here. What distance to the coast do these results refer to?Even in coastal areas shipping is seldom the main contributing source, almost always surpassed by traffic contributions (see for Spain the works by Pandolfi et al., Pérez et al, iana et al., Amato et al....). It seems unlikely that shipping accounts for 50% of NO2 ambient concentrations inland. Or are the authors referring to air emissions? If so, this could be possible for major cities such as Barcelona. Please clarify the meaning of these sentences.

- line 249: once again, model validation is needed here.

- lines 265-268 and 283-284: please remove the references to the "port of", as the model's resolution is too coarse to capture this.

- section 3.3, please add model validation and the issue of primary and secondary aerosols, as limitations.

- line 333, suggestion to add reference to the HEI Special Report

---

## Referee Comment (RC2) · Anonymous Referee #2 · 30 Mar 2020

The study evaluates the impacts of shipping emissions on the air quality in the region of the Iberian Peninsula and the Strait of Gibraltar, one of the busiest maritime routes in the world. This chemistry-transport modelling study makes use of shipping emissions generated by the STEAM 3 model that allocates ship activities via the Automatic Identification System operating onboard the vessels. Among the valuable information presented in this manuscript are a comparison of ship emission intensities with those reported for ports in the Asian region and a calculation of the ship impact on exceedances of regulatory air quality limits. Unfortunately, it is not immediately apparent what the manuscript adds to already published chemistry transport modelling studies on the impact of ship emissions in Europe. Overall, the manuscript reads more like a good technical report than a research article, as the applied methods are not originally

proposed and the uncertainties of model results are not comprehensively discussed and quantified.

The spatial resolution is the same as in prior studies that covered the whole of Europe. This is somewhat surprising, given that a prior study by Monteiro et al. (2018) in the same region used a finer resolution (3 km x 3km). The applied difference method for quantifying the shipping emission contribution is flawed since the effect of nonlinearities in ozone chemistry on the ship impact was not evaluated, despite the high photochemical activity in this region. For both daytime and nighttime, the instantaneous NOx lifetime in ship plumes is a strong function of the initial NOx concentration at ship stack, resulting in a very nonlinear loss rate for NOx in ship plumes (e.g. Song et al., 2003). Model procedures that shift ship plume levels by an order of magnitude, as can be expected for a 10-km wide grid cell, will quite likely overestimate NOx lifetime.

The heat release from ship stack exhaust of large ships represents a buoyancy flux that may result in plume rise. Therefore, we can expect that a significant fraction of the shipping emissions are emitted at upper heights. The STEAM 3 model should be able to take into account plume rise of ship exhaust in generalized form. A description of the treatment of the vertical distribution of shipping emissions and injection heights that are used for the corresponding vertical layers of the EMEP modelling system should be added to the method section. When shipping emissions have been fully transferred to the lowest vertical model layer, such a procedure has to be justified and the error due to this needs to be approximated.

The significance of the modelled ship contribution was not validated with measurements. Although the Norwegian Meteorological Institute regularly validates the air quality predictions with the EMEP MSC-W model for Europe, it is not sufficient to simply refer to this. The manuscript should include a validation of the modelled concentrations in the subdomain region with monitoring data from stations in Portugal, Spain and France for 2015 (EMEP network, EEA AirBase, EBAS database). The comparison should include model data from both runs with and without shipping emissions.

Specific Comments:

1.) P. 1 lines 24 - 26: Many studies can be found about the impacts of shipping emissions on air quality and health. It would be a good place here to discuss deviations and contradictions in the literature concerning the relevance of shipping for health impacts, and specifically the roles of primary versus secondary particulate matter.

2.) P. 4 lines 111 - 114: Which boundary conditions of the chemical concentrations were used for the subdomain runs?

3.) P. 5 line 157 to P. 6 line 173: Suggest to transfer the information of annual average emission intensities (per pollutant and per port/sea area) into a table to facilitate the comparison with shipping activity in the Asian region.

4.) Impact on Air Quality: Suggest to divide section 3.2 in topical subsections; for example "Annual average concentrations" (P. 7 lines 202 - 218), "Comparison with previous studies in the region" (P. 7 line 219 to P. 8 line 250), "Seasonal variation" (P. 8 lines 251 - 260), "Possible health impacts" (P. 8 line 261 to P.9 line 297). Some passages could be shortened.

5.) P. 9 lines 276 - 287: Suggest to illustrate the contribution of shipping emissions to the exceedances of limit values in form of a bar diagram, i.e. showing the increment of number of exceedances (NO2, PM2.5, PM10) and number of days of exceedances (SO2) due to ship traffic for the major ports of the Iberian Peninsula.

6.) Uncertainties and limitations: The uncertainties of the emission factors of pollutants from different ship types could easily dominate the uncertainty of the evaluated contribution from shipping. With the STEAM 3 model at hand, it should be possible to estimate the overall uncertainty in the modelled concentrations due to uncertain emission factors. To arrive at a more reliable margin of the contribution of shipping emissions in this region, my request to the authors is that they perform shipping emission calculations with the respective lower and upper bound of the emission factors of

NOx, SOx and primary particulates, then repeating the runs with EMEP MSC-W using the lower and higher emission dataset.

Technical Corrections:

P. 1 lines 17 - 18: "ktonnes y-1" is not a SI unit.

P. 1 line 27: on a global scale?

P. 1 lines29 - 30: reference(s) for this statement missing.

P. 3 lines 66 - 67: suggest to reference the study of Ramacher et al. (2019) on local scale for Baltic Sea ports.

P. 5 line 150: "ash" – what is this chemically? Please define.

P. 8 line 232: please replace "lower increases contributions" by "lower positive contributions".

Conclusions: the word "verify" is used several times in the conclusions section (P. 10, line 321; P. 10, line 324; P. 11, line 330). Verification implies the comparison of model results to the true values, which are not known. Please change wording.

P. 11 line 340: what about the code availability of STEAM 3? Please include a statement here.

P. 15 lines 469-471: the citation of Marelle et al. is incomplete.

Table 1 and Table 2: "tonne y-1" is not SI unit.

Figure 1 and Figure 2: please use SI units in labels, axis annotations and captions.

Figure 4f: what is the cause for high O3 values along the North African coast over water?

References:

Ramacher, M. O. P., Karl, M., Bieser, J., Jalkanen, J.-P., and Johansson, L.:

Urban population exposure to NOx emissions from local shipping in three Baltic Sea harbor cities – a generic approach, Atmos. Chem. Phys., 19, 9153–9179, https://doi.org/10.5194/acp-19-9153-2019, 2019.

Song, C. H., Chen, G., Hanna, S. R., Crawford, J., and Davis, D. D.: Dispersion and chemical evolution of ship plumes in the marine boundary layer: Investigation of O3/NOy/HOx chemistry, J. Geophys. Res., 108 (D4), 4143, doi:10.1029/2002JD002216, 2003.

---

## Referee Comment (RC3) · Anonymous Referee #3 · 31 Mar 2020

In their manuscript "Shipping emissions in the Iberian Peninsula and its impacts on air Quality", Rafael Nunes and colleagues report on a study investigating the magnitude and effects of shipping emissions around the Iberian Peninsula. First, they use the STEAM3 model to estimate ship emissions for a number of relevant quantities. Then, they use the EMEP MSC-W model to determine the impact of these emissions on surface concentrations of the trace gases and aerosols they consider. Finally, they determine for all model cells in the domain whether or not the inclusion of the shipping emissions leads to additional exceedances of the WHO and EU air quality guidelines.

The manuscript is clear and well written and fits into the scope of ACP. The setup of the study is logical and straight forward and the results are interesting for anyone modelling air quality for the Iberian Peninsula. Although there is nothing really new about

the methods and approach used, and there are no surprises in the results, the paper still contributes to our body of knowledge on pollution and air quality. I therefore recommend it for publication in ACP after the points listed below have been satisfactorily addressed.

**Major comment**

My main criticism of the manuscript is that the results on additional exceedances of air quality standards due to ship emissions (which are potentially of interest to policy makers) depend strongly on the quality of the modelled fields. Only if they give a good representation of the actual air quality and exceedances, then the difference between the results with and without ship emissions can be trusted. I therefore believe that the authors need to include a comparison of the modelled concentrations and exceedances for the scenario including ship emissions to those measured by in-situ air quality networks to demonstrate that they are close enough to reality to make interpretation of delta exceedances worthwhile.

**Minor comments**

- While the manuscript is overall well written, it would benefit from proof reading by a native speaker

- page 4, line 120: Are Sahara dust emissions and NOx from lightning really taken from the NCAR fire inventory?

- page 7, line 204 and figures: I think it is stated nowhere that when you talk about concentrations, that always means at the surface (I assume)

- page 8, line 251 and following: I'm a bit confused by this discussion of the origins of the seasonality. It sounds as if it is not really clear what the origin is, but don't you have all the information on the magnitude of emissions from STEAM so that you can give clear answers on what drives the seasonality?

- page 10, line 299: The discussion on uncertainties and limitations is very general indeed and mainly lists the obvious. I think that the comparison to real data will make this section also more relevant.

- Figure 1: I'm not sure that it makes really sense to show all these figures here – they all look the same with the colour scale chosen and I do not see what I can learn from 8 figures which I cannot already see in the first.

---

## Author Response (AR1)

Dear Editor,

Please fin attached the point-by-point response to the reviews.

Thank you.

Kind regards,

Sofia Sousa

**AUTHOR'S RESPONSES TO REFEREE #1:**

We thank Referee #1 for positive evaluation and for taking the time to read and give valuable comments to improve the manuscript. Following the reviewer remarks, we addressed the comments and questions in detail.

1)"Model validation: no evidence is preported that the model was validated with observations. Was it validated in any way? How? Even if it cannot be validated with actual shipping contributions due to the different methods used (lines 237-238), the authors could compare their results to total NO2 or other gaseous pollutant concentrations from reference stations, for example. This kind of comparison would be essential to confirm their modelling results."

Answer: Thank you for your comments. Although information about model validation can be found in lines 129-134: "Regarding the performance of the model, simulations from EMEP/MSC-W are regularly evaluated against measurements in the EMEP annual reports (Norwegian Meteorological Institute, 2018). Moreover, there are several studies that compare model results with measurements and calculations with other models (Angelbratt et al., 2011; Bessagnet et al., 2016; Colette et al., 2011, 2012; Jonson et al., 2010; Karl et al., 2017; Prank et al., 2016; Soares et al., 2016) and recent studies that used the model to assess the effects of shipping emissions (Jonson et al., 2015, 2017; Turner et al., 2017)", in order to support our results, model output  $PM_{2.5}$ ,  $PM_{10}$  and  $NO_2$  concentrations for the S-SCN scenario were compared with data from the monitoring stations of EU Member States reported by the European Environmental Agency for 2015. Moreover, comparisons between the modelling reference results reported by EMEP for the year 2015 were also compared with the data from the monitoring stations. Annual mean concentrations observed in 139 stations for PM2.5, 337 stations for PM10 and 446 stations for NO2 were compared with the model results in time and space. Information about model validation will be added in the Methods section as follows: "...and recent studies that used the model to assess the effects of shipping emissions (Jonson et al., 2015, 2017; Turner et al., 2017). To support the results of the present study, model output  $PM_{2.5}$ ,  $PM_{10}$  and  $NO_2$ concentrations for the S-SCN scenario were compared with data from the monitoring stations of EU Member States reported by the European Environmental Agency for 2015 (EEA, 2020). Moreover, comparisons between the modelling reference results reported by EMEP for the year 2015 (Norwegian Meteorological Institute, 2019) were also compared with the data from the monitoring stations. Annual mean concentrations observed in 139 stations for PM2.5, 337 stations for  $PM_{10}$  and 446 stations for NO2 were compared with the model results in time and space. Table 1 summarizes the model quality indicators (Pearson correlation coefficient (Pearson's r), Mean Bias Error (MBE), Mean Absolute Error (MAE) and Root Mean Square Error (RMSE)), for the present study and for the reference results reported by EMEP. Similar results were obtained for the comparison with the present study and with the reference results of EMEP, which indicates that the model simulations were well executed. Correlations obtained were moderately positive (Pearson's r > 0.5) for all pollutants, with errors smaller than those reported in the literature (Monteiro et al., 2018)."

| Indicators        |                   | This study       |                 | E۸                | AEP reference    | ce              |
|-------------------|-------------------|------------------|-----------------|-------------------|------------------|-----------------|
|                   | PM 2.5 | PM 10 | NO 2 | PM 2.5 | PM 10 | NO 2 |
| Pearson's r       | 0.57              | 0.55             | 0.70            | 0.64              | 0.55             | 0.67            |
| MBE a  | 1.32              | 19.51            | 5.78            | 0.34              | 18.70            | 5.19            |
| MAE b  | 2.86              | 19.55            | 8.70            | 2.81              | 18.74            | 9.18            |
| RMSE c | 3.62              | 20.83            | 11.24           | 3.59              | 20.11            | 11.90           |

 Table 1. Model quality indicator values for the present study and for the reference results

 reported by EMEP.

a Mean Bias Error; b Mean Absolute Error; c Root Mean Square Error

2) Primary vs. secondary aerosol contributions: it is unclear in the manuscript whether the particle concentrations modelled are primary or primary + secondary aerossol from shipping. If secondary aerosols were included, how was this implemented in the model? This is the main limitation of most models targeting shipping emissions (both dispersion and receptor models). Please address this carefully in the Methods section.

Answer: Thank you for your comments. The secondary aerosols were included in the model. In the EMEP MSC-W model  $PM_{2.5}$  concentrations were defined as  $PM_{2.5} = SO_4^{2-} + NO_3^-$  (fine) +  $NH_4^+$  + SS(fine) +  $PPM_{2.5} + 0.27 NO_3^-$  (coarse) considering the secondary organic aerosols as the aerosol mass arising from the oxidation products of gas-phase species, the secondary inorganic aerosols as  $SO_4^{2-} + NO_3^-$  (fine) +  $NH_4^+ + NO_3^-$  (coarse), sea salt (SS) and the primary particulate matter ( $PPM_{2.5}$  and PPMcoarse) originating directly from anthropogenic emissions (as was the case of shipping emissions).  $PM_{10}$  concentrations were calculated as  $PM_{10} = PM_{2.5}+PM$ coarse where PMcoarse was defined as PMcoarse =  $0.33 NO_3^-$  (coarse) + SS(coarse) + PPMcoarse. Information about how PM concentrations were modelled in this study will be added in the Methods section as follows: "... having a thickness of 50 m. PM concentrations were modelled considering primary particulate matter originating directly from anthropogenic emissions, as well as secondary organic and inorganic aerosols and sea salt. Other details about the model can be found in Simpson et al. (2012) and in Norwegian Meteorological Institute (2017a)."

**Specific comments:**

- line 25, "its contribution", does this refer to health impacts? The contribution to air quality degradation has been assessed in numerous papers in the literature, including the papers referenced by the authors.

Answer: Yes, we were referring to the contribution for human health degradation. We decided to change to: "... which may lead to known negative effects on air quality and health, being its contribution to human health degradation still not well documented (Brandt et al., 2013; Corbett et al., 2007; Nunes et al., 2017b; Sofiev et al., 2018)."

**- the English could be reviewed by a native speaker, it is good but some small typos remain.**

Answer: Suggestion attended. The manuscript will be review by a native speaker.

- line 33, suggestion to reference the EEA report EEA, 2013. The impact of international shipping on European air quality and climate forcing. EEA Technical Report 04/2013. Luxembourg: Publications Office of the European Union, 2013. ISBN 978-92-9213-357-3.

Answer: Suggestion attended. The reference will be added.

- Please add in the Methods section discussions on model validation and on secondary aerosols (whether they are or not included in the model).

Answer: Suggestion attended. More information about model validation will be added, according to our previous answers.

**- line 150, what does "ash" refer to, exactly? Please define**

Answer: Thanks for your comment. Once in STEAM PM emissions were calculated as the sum of SO4, H2O, EC, OC and ash, considering the different emission factors, we chose to maintain this separation. Ash refers to a component of the PM emitted by ships and depends on the content of marine fuels. To give more information about the ash component and emission factors used in STEAM we will add the following sentence: "... sulphates and ash (a component of the PM emitted by ships that depends on the content of marine fuels) for the Iberian Peninsula in 2015 in a 0.1°x0.1° grid cells (approximately 10 x 10 km2). Details about emission factors used in STEAM can be found in Jalkanen et al. (2009), Jalkanen et al. (2012) and Jonson et al. (2014)."

- line 155, "ports", the resolution is quite coarse (10x10 km2) to represent harbour emissions, or even most coastal urban areas. Please highlight this as a limitation.

Answer: Thanks for your comment. Our objective was not to make a detailed analysis of emissions or concentrations in ports. Despite the limitation of the grid used (10x10km), it was possible to identify higher emissions for the cells near the port areas. Anyway, we will add the fact that this resolution is too coarse to make a detailed analysis of emissions and concentrations in ports as a limitation of the study in the "Uncertainties and limitations"

section as follows: "...Furthermore, EMEP-MSC/W model has been recently compared with the CMAQ and the SILAM models and showed the best spatial correlation of annual mean concentrations for NO2, SO2 and PM2.5 resulting of shipping emissions, although it seems to be underestimating PM2.5 concentrations and overestimating O3 concentrations. Moreover, although it has been possible to identify variations in the emissions and concentrations near the port areas, the resolution that was used was too coarse to make a detailed analysis of emissions and concentrations inside the port areas."

**- line 160, suggest to check and reference the report HEI Special report 22, Impacts of shipping on air pollution emissions, air quality, and health in the Yangtze River Delta and Shanghai, China**

Answer: Thank you for your comment. We checked the results of the report HEI Special report 22 and information about the differences in the emissions intensities will be added as follows: "Nevertheless, in the HEI report authors described lower emission intensities for the Yangtze River Delta and Shanghai areas at 12 NM from the coast. According to these results, comparisons should be made carefully as emission intensities seem strongly dependent on the location for which they are calculated (inside the port area, at a certain distance from the coast or on the high seas) and also on the methodology used to calculating shipping emissions."

**- lines 185-188, please add a statistical trend analysis: the differences don't seem statistically significant, to the naked eye.**

Answer: Thank you for your comment. Statistical trend analysis will be added. The ranked nonparametric test Mann-Kendall trend test was used for detecting monotonic trends in the monthly emissions. The null hypothesis H0 was assumed as "there is no trend in the emissions over the months" and it was tested against the alternative hypothesis H1 which considered that "there is increasing or decreasing trend in the monthly emissions". The tests performed at the 95% confidence interval level showed no statistically significant trends in the monthly emissions data. Information about statistical trend analysis will be added as following: "It can be observed that emissions increased progressively from February to July, where they reached the maximum annual value. After that, a decrease during August and September was observed, followed by a stabilization during October (for some pollutants there was a slight increase) and a decrease until December. Although emissions varied throughout the year, variations were about 1-2% between months and each month represented 7.1-9.1% of the annual total emissions. In fact, according to the statistical trend analysis using the Mann-Kendall trend test, performed at the 95% confidence interval level, no statistically significant variations were achieved in the monthly emissions data for all pollutants (*p*-values > 0.05)."

- line 191: I don't think the comparison with a paper from 1999 (even if a reference paper) is adequate here: in 20 years the trade and sailing patterns have surely changed largely, therefore this comparison is not representative. Answer: Suggestion attended. This information will be deleted from the manuscript.

**- line 203, are these primary or secondary PM10 and PM2.5 concentrations? Or the sum of both?**

Answer: These are primary and secondary  $PM_{10}$  and  $PM_{2.5}$ . Details about this issue were already described in a previous answer.

- line 206, why do concentrations increase gradually towards and over the N of Africa? Are there no O3 sinks (e.g., major cities) in this region? This seems unlikely, probably the emission inventories are not accurate for this region. Please discuss.

Answer: Thanks for your comment. Actually, it is not over the North of Africa but close to it, near the coast, thus on the sea area, and not over the region. That is why there are no major cities there.

- line 217, are these (4.8 microg/me and 6.9 microg/m3) shipping contributions? They seem quite high, especially if only primary aerosols are considered (I'm still unsure of this). Also, are these average values for the entire peninsula? Please compare with shipping contributions from the literature, and also with total (non-ship sourced) PM10, NO2, O3, etc concentrations.

Answer: Thank you for your comments. As already mentioned primary and secondary aerosols were considered for modelled PM concentrations. These values are not average values, but maximum values that were verified in one grid cell of the domain. Comparisons with average values were performed with other studies and are in lines 220-224.

- line 240-241: these contributions seem unlikely as they are reported here. What distance to the coast do these results refer to? Even in coastal areas shipping is seldom the main contributing source, almost always surpassed by traffic contributions (see for Spain the works by Pandolfi et al., Pérez et al, iana et al., Amato et al....). It seems unlikely that shipping accounts for 50% of NO2 ambient concentrations inland. Or are the authors referring to air emissions? If so, this could be possible for major cities such as Barcelona. Please clarify the meaning of these sentences.

Answer: Thank you for your comments. These results are referring to the contribution of ship emissions to annual mean concentrations calculated as  $[((S-SCN) - (B-SCN)) / (B-SCN)] \times 100$ . Moreover, these results (the 50% contribution) refer to inland zones close to the biggest port areas (as can be seen from Figure 4 a)). It was also possible to identify contribution of around 75% for inland regions close to the Strait of Gibraltar.

- line 249: once again, model validation is needed here.

Answer: Thank you for your comment. Model validation will be added as follows: "Monteiro et al. (2018) reported for the west coast of Portugal (also the west coast of Iberian Peninsula) lower contributions for NO2 and PM10 (higher than 20% and less than 5%, respectively) than those reported in this study probably due to the different methodology applied. Moreover, according to the model validation made by Monteiro et al. (2018), their model underestimated PM10 and NO2 concentrations (negative MBE), while the model used in the present study overestimated them (positive MBE)."

**- lines 265-268 and 283-284: please remove the references to the "port of", as the model's resolution is too coarse to capture this.**

Answer: Thank you for your comment. We will change to "area close of Port".

**- section 3.3, please add model validation and the issue of primary and secondary aerosols, as limitations.**

Answer: Thank you for your comment. Model validation will be added as above described. Primary and secondary aerosols were considered in the model, thus this is not a limitation of the study.

**- line 333, suggestion to add reference to the HEI Special Report**

Answer: Suggestion attended. The reference will be added.

**AUTHOR'S RESPONSES TO REFEREE #2:**

We thank Referee #2 for the positive evaluation and for taking the time to read our paper and giving us valuable comments to improve the manuscript. Following the reviewer remarks, we addressed the comments and questions in detail below.

1)"The study evaluates the impacts of shipping emissions on the air quality in the region of the Iberian Peninsula and the Strait of Gibraltar, one of the busiest maritime routes in the world. This chemistry-transport modelling study makes use of shipping emissions generated by the STEAM 3 model that allocates ship activities via the Automatic Identification System operating on board the vessels. Among the valuable information presented in this manuscript are a comparison of ship emission intensities with those reported for ports in the Asian region and a calculation of the ship impact on exceedances of regulatory air quality limits. Unfortunately, it is not immediately apparent what the manuscript adds to already published chemistry transport modelling studies on the impact of ship emissions in Europe. Overall, the manuscript reads more like a good technical report than a research article, as the applied methods are not originally proposed and the uncertainties of model results are not comprehensively discussed and quantified."

Answer: Thank you for your comments. With this study, we try to give an overall view of the air quality impact of the shipping emissions over the Iberian Peninsula (specifically) using the STEAM, that is considered one of the most reliable methods to estimate shipping emissions (exhibits the highest spatially resolution in their emissions and a large number of secondary routes that do not appear in other inventories), and EMEP for the air quality modelling. Only another study estimated the concentrations for the Iberian Peninsula, but used different methodologies to do it (different inventory and CTM, referred in lines 69-78).

Furthermore, this is the first time that the modelling results of this new version of STEAM are discussed in detail for the Iberian Peninsula (this inventory was already discussed but from a global point of view, not specifically for the Iberian Peninsula) and the very first time that they were used to estimate the pollutant concentrations for the Iberian Peninsula.

Moreover, to support our results a model validation will be added. Model output  $PM_{2.5}$ ,  $PM_{10}$  and  $NO_2$  concentrations for the S-SCN scenario were compared with data from the monitoring stations of the EU Member States reported by the European Environmental Agency for 2015. Moreover, comparisons between the modelling reference results reported by EMEP for the year 2015 were also compared with the data from the monitoring stations. Annual mean concentrations observed in 139 stations for  $PM_{2.5}$ , 337 stations for  $PM_{10}$  and 446 stations for  $NO_2$

were compared with the model results in time and space. Also, a comparison between the exceedances for the scenario including ship emissions and the levels from the monitoring stations was also made. We estimate the percentage of exceedances that our model found in relation to the exceedances detected with the concentrations of the monitoring stations. More information was added to the Uncertainties chapter to improve the discussion.

2) "The spatial resolution is the same as in prior studies that covered the whole of Europe. This is somewhat surprising, given that a prior study by Monteiro et al. (2018) in the same region used a finer resolution (3 km x 3km). The applied difference method for quantifying the shipping emission contribution is flawed since the effect of nonlinearities in ozone chemistry on the ship impact was not evaluated, despite the high photochemical activity in this region. For both daytime and night time, the instantaneous NOx lifetime in ship plumes is a strong function of the initial NOx concentration at ship stack, resulting in a very nonlinear loss rate for NOx in ship plumes (e.g. Song et al., 2003). Model procedures that shift ship plume levels by an order of magnitude, as can be expected for a 10-km wide grid cell, will quite likely overestimate NOx lifetime."

Answer: Thank you for your comments. As shipping is a major source of NOx to the troposphere, and especially because large amounts of these pollutants are often released from point sources into the relatively clean maritime atmosphere, since the version rv4.8 of the EMEP MSC-W a new pseudo-species "ShipNOx" has been introduced and NOx released by ships started to be treated differently and not like any other source of NOx. Like you said in your comment, in 3-D models NOx emissions are typically diluted into large grid volumes which can lead to large over-predictions in O3 production, and in the NOx lifetime. Tests with the EMEP model confirmed these issues and also that the early approach was not appropriate for the European area at least (considered ship-related NOx emissions like any other source of NOx). To prevent such effects, the model assigns 50% of shipping NOx to the pseudo-species ShipNOx and the rest as NO and NO2 as previously done. ShipNOx deposits as NO2, but suffer simple atmospheric reactions:

ShipNOx + OH $\Rightarrow$ HNO3 Reaction 1; ShipNOx $\Rightarrow$ HNO3 Reaction 2

Reaction 1 proceeds with the same rate as the normal NO2+ OH reaction, thus proceeding faster in daylight and in areas with high-OH. Reaction 2 provides a minimum half-life of about 6 hours, accordingly to Vinken et al. (2011) results.

The heat release from ship stack exhaust of large ships represents a buoyancy flux that may result in plume rise. Therefore, we can expect that a significant fraction of the shipping emissions are emitted at upper heights. The STEAM 3 model should be able to take into account plume rise of ship exhaust in generalized form. A description of the treatment of the vertical distribution of shipping emissions and injection heights that are used for the corresponding vertical layers of the EMEP modelling system should be added to the method section. When shipping emissions have been fully transferred to the lowest vertical model layer, such a procedure has to be justified and the error due to this needs to be approximated.

Answer: Thank you for your comments. The estimated height distribution of emissions from STEAM are given in the figure below. According to these estimates, over 80% of emissions occur between 30-60 m height. The plume release height is a function of vessel type, size and plume rise. For the two former, vessel stack height is determined from photographs and existing data from IHS Markit. However, the airdraft (how high the vessel is from the water surface) or the keel-to-mast height are rarely available in commercially available databases. For this reason, we have used vessel scale drawings and photographs from Significant Ships publication serie, to link vessel types and sizes to stack height. There is a linear dependency between vessel stack height and vessel length, but these linear functions are vessel type specific. Regardless, STEAM does not consider plume rise, because exhaust temperature, exhaust velocity and funnel pipe diameter are not known. In principle, some typical values could be used, but currently STEAM only adds a constant value of 10m to stack height estimation to provide a primitive estimate of the plume rise.

The detailed height profile for emissions obviously depends on the type of traffic operating in the area. Largest cruise vessels can have very high stacks, exceeding 70m height and plume rise may elevate the plumes even higher. However, STEAM does not currently consider meteorology during emission calculation. In our opinion, the plume rise issue in ship exhaust dispersion is most relevant for local scale air quality assessments, but less so for regional scale work. For this reason, the ship emissions from STEAM were allocated to the first model layer of the EMEP runs. Information related to this issue will be added to the Methods section as follows: "... having a thickness of 50 m. Assuming that the plume rise issue in ship exhaust dispersion is

more relevant for local scale air quality assessments, and less for regional scale work. For this reason, the ship emissions from STEAM were allocated to the first model layer of the EMEP runs."

"The significance of the modelled ship contribution was not validated with measurements. Although the Norwegian Meteorological Institute regularly validates the air quality predictions with the EMEP MSC-W model for Europe, it is not sufficient to simply refer to this. The manuscript should include a validation of the modelled concentrations in the subdomain region with monitoring data from stations in Portugal, Spain and France for 2015 (EMEP network, EEA AirBase, EBAS database). The comparison should include model data from both runs with and without shipping emissions."

Answer: Thank you for your comments. To support our results, model output PM2.5, PM10 and  $NO_2$  concentrations for the S-SCN scenario were compared with data from the monitoring stations of the EU Member States reported by the European Environmental Agency for 2015. Moreover, comparisons between the modelling reference results reported by EMEP for the year 2015 were also compared with the data from the monitoring stations. Annual mean concentrations observed in 139 stations for  $PM_{2.5}$ , 337 stations for  $PM_{10}$  and 446 stations for  $NO_2$ were compared with the model results in time and space. Information about model validation will be added in the Methods section as follows: "...and recent studies that used the model to assess the effects of shipping emissions (Jonson et al., 2015, 2017; Turner et al., 2017). To support the results of the present study, model output  $PM_{2.5}$ ,  $PM_{10}$  and  $NO_2$  concentrations for the S-SCN scenario were compared with data from the monitoring stations of the EU Member States reported by the European Environmental Agency for 2015 (EEA, 2020). Moreover, comparisons between the modelling reference results reported by EMEP for the year 2015 (Norwegian Meteorological Institute, 2019) were also compared with the data from the monitoring stations. Annual mean concentrations observed in 139 stations for PM2.5, 337 stations for  $PM_{10}$  and 446 stations for  $NO_2$  were compared with the model results in time and space. Table 1 summarizes the model quality indicators (Pearson correlation coefficient (Pearson's r), Mean Bias Error (MBE), Mean Absolute Error (MAE) and Root Mean Square Error (RMSE)), for the present study estimations and for the reference results reported by EMEP. Similar quality indicators were obtained for the comparison the results of the present study and the reference results of EMEP, which indicates that the model simulations were well executed. Although the correlations obtained were moderate positive correlations (Pearson's r > 0.5) for all pollutants, the errors obtained were smaller than those reported in the literature (Monteiro et al., 2018), which makes our results acceptable."

**Table 1.** Model quality indicators for the present study estimations and for the reference resultsreported by EMEP.

| Indicators        |                   | This study       |                 | E۸                | AEP reference    | ce              |
|-------------------|-------------------|------------------|-----------------|-------------------|------------------|-----------------|
|                   | PM 2.5 | PM 10 | NO 2 | PM 2.5 | PM 10 | NO 2 |
| Pearson's r       | 0.57              | 0.55             | 0.70            | 0.64              | 0.55             | 0.67            |
| MBE a  | 1.32              | 19.51            | 5.78            | 0.34              | 18.70            | 5.19            |
| MAE b  | 2.86              | 19.55            | 8.70            | 2.81              | 18.74            | 9.18            |
| RMSE c | 3.62              | 20.83            | 11.24           | 3.59              | 20.11            | 11.90           |

a Mean Bias Error; b Mean Absolute Error; c Root Mean Square Error

**Specific Comments:**

1.) P.1 lines 24-26: Many studies can be found about the impacts of shipping emissions on air quality and health. It would be a good place here to discuss deviations and contradictions in the literature concerning the relevance of shipping for health impacts, and specifically the roles of primary versus secondary particulate matter.

Answer: Suggestion attended. We will change the following sentences: "Marine traffic has been identified as a relevant source of pollutants especially nitrogen oxides ( $NO_x$ ), sulphur oxides ( $SO_x$ ) and particulate matter (PM), which may lead to known negative effects on air quality and health, being its contribution to human health degradation still not well documented (Brandt et al., 2013; Corbett et al., 2007; Nunes et al., 2017b; Sofiev et al., 2018). Studies have been reporting that shipping contributions to ambient PM in port areas are mainly secondary particles (around 60 to 70% of PM10 and PM2.5 mass concentrations). Despite this, studies have been suggesting that could be more advantageous to reduce shipping-related primary particle emissions than precursors of secondary particles ( $NO_x$  and  $SO_x$ ), which are the target of current international regulations (Viana et al., 2014)."

**2.) P.4 lines 111-114: Which boundary conditions of the chemical concentrations were used for the subdomain runs?**

Answer: Information about boundary conditions used in the chemical transport model for the subdomain runs will be added as: "...400 km from the Iberian Peninsula coast. The initial and the lateral boundary conditions for most of the chemical compounds were defined by functions defining concentrations in terms of latitude and time, based on measurements and/or model calculations, providing robustness which chemical transport model results sometimes lack. More information about the EMEP/MSC-W configuration for initial and boundary concentrations used in this study can be found in Simpson et al. (2012)."

**3.) P. 5 line 157 to P. 6 line 173: Suggest to transfer the information of annual average emission intensities (per pollutant and per port/sea area) into a table to facilitate the comparison with shipping activity in the Asian region.**

Answer: Suggestion attended. We will change the following sentences: "The annual average intensities of ash, CO, CO2, EC, NOx, OC, sulphate and SOx emissions were 9.0E-04 tonnes/yr/km2, tonnes/yr/km2, 1.38E-02 8.47 tonnes/yr/km2, 1.27E-01 tonnes/yr/km2, 1.97E-01 tonnes/yr/km2, 3.16E-03 tonnes/yr/km2, 8.04E-03 tonnes/yr/km2 and 1.01E-01 tonnes/yr/km2, respectively. The annual average and highest intensities for NOx and SOx reported for the Asian Region are present in Table 3 (Chen et al., 2016a, 2017; Fan et al., 2016). In general, the average intensities that were reported for Asia were considerably higher than those found in this study. It was possible to identify in the present study two main hubs given the high emissions intensity: Valencia Port and the Strait of Gibraltar. At Valencia Port, ash, CO, EC and OC had the highest values, respectively, 1.46E-01 tonnes/yr/km2, 1.85 tonnes/yr/km2, 1.99E-01 tonnes/yr/km2 and 5.09E-01 tonnes/yr/km2. At the Strait of Gibraltar,  $CO_2$ ,  $NO_x$ , sulphate and  $SO_x$  had the highest values, respectively, 1330 tonnes/yr/km2, 24 tonnes/yr/km2, 1.03 tonnes/yr/km2 and 11.6 tonnes/yr/km2. In accordance to what was referred above, in the Asian Region maxima intensities were also higher than those here estimated (Chen et al., 2016b; Fan et al., 2016; Ng et al., 2013). "We will also add Table 3.

| Study               |                | NO x |                  | SO x |                  |  |
|---------------------|----------------|-----------------|------------------|-----------------|------------------|--|
|                     | Port/sea area  | Annual average  | Highest
value | Annual average  | Highest
value |  |
| Chen et al. (2016a) | Tianjin Port   | 5.06            | 1.51E+03         | 7.14            | 1.79E+03         |  |
| Chen et al. (2017)  | Qingdao Port   | 1.83            | -                | 1.42            | -                |  |
| Fan et al. (2016)   | East China Sea | 1.0             | 1.0E+04          | 1.90            | 1.30E+03         |  |
| Ng et al. (2013)    | Hong Kong      | -               | 1.1E+02          |                 | 2.0E+02          |  |

**Table 3.** Annual average and highest intensities of  $NO_x$  and  $SO_x$  (in tonnes/yr/km2) reported from researches in Asian Region.

4.) Impact on Air Quality: Suggest to divide section 3.2 in topical subsections; for example "Annual average concentrations" (P. 7 lines 202 - 218), "Comparison with previous studies in the region" (P. 7 line 219 to P. 8 line 250), "Seasonal variation"

**(P.8 lines 251 - 260), "Possible health impacts" (P. 8 line 261 to P.9 line 297). Some passages could be shortened.**

Answer: Suggestion attended. We will subdivide the section 3.2.

5.) P. 9 lines 276-287: Suggest to illustrate the contribution of shipping emissions to the exceedances of limit values in form of a bar diagram, i.e. showing the increment of number of exceedances (NO2, PM2.5, PM10) and number of days of exceedances(SO2) due to ship traffic for the major ports of the Iberian Peninsula.

Answer: Thank you for your comment. We believe that with a bar diagram the spatial distribution (illustrated with Figure A2) will be lost.

6.) Uncertainties and limitations: The uncertainties of the emission factors of pollutants from different ship types could easily dominate the uncertainty of the evaluated contribution from shipping. With the STEAM 3 model at hand, it should be possible to estimate the overall uncertainty in the modelled concentrations due to uncertain emission factors. To arrive at a more reliable margin of the contribution of shipping emissions in this region, my request to the authors is that they perform shipping emission calculations with the respective lower and upper bound of the emission factors of NOx, SOx and primary particulates, then repeating the runs with EMEP MSC-W using the lower and higher emission dataset.

Answer: Thank you for your comments. In STEAM, there are several sources of uncertainty which can have an impact on the accuracy of the results. These could be classified in three categories:

- a) Gaps in input data (incomplete AIS coverage, missing IHS Markit data)
- b) Power prediction (weather contributions, Hollenbach resistance inaccuracy, fouling, squat, sea currents, aux engine power profiles, engine load estimation, power transmission, propeller properties)
- c) Emission factors (specific fuel oil consumption, fuel type, fuel sulphur content allocation, engine generation)

Each of these three categories can have multiple contributions, indicated by various error sources in parenthesis. STEAM has mechanisms to mitigate most of the uncertainties listed above and some are features, like weather, are currently developed, but will be reported separately at a later stage. Uncertainties concerning emission factors may be larger for products of incomplete combustion, like CO, NMVOC, OC and EC, than for  $CO_2$ , or  $NO_x$ , because these are strongly related to engine load, engine generation and service history. The emission factors may also depend on the fuel type assignment and fuel sulphur content, which are

estimated based on engine properties and maximum sulphur content allowed in each region at the time period of the study. However, the emission factors for incomplete combustion products may be affected by engine service history and thus are notoriously difficult to estimate. We are not currently aware of any study which would provide uncertainty evaluations for all emission sectors and emission factors included in regional air quality modelling and it seems curious to us to demand one for shipping, only. To conduct such a study would require many computer simulations and significant additional effort. Even if these tests would be limited to uncertainty evaluations of three air pollutants modelled by STEAM, it would still require low- and high-bound runs with STEAM and consecutive analysis with the EMEP model. Even then, the uncertainty evaluation would not be just about emission factors, because primary particulates would also require adjusting the assumptions concerning how the fuel sulphur content was assigned and what are its consequences on PM components. This will multiply the work required by at least a factor of six, which is not currently possible due to limitations in available research funding.

**Technical Corrections:**

**P. 1 lines 17-18: "ktonnes y-1" is not a SI unit.**

Answer: Given that the values are high, it is usual to present the results in above referred units. We maintained the units to be more coherent with the literature.

**P. 1 line 27: on a global scale?**

Answer: Suggestion attended. Yes, it is on a global scale. This information will be added.

**P. 1 lines29-30: reference(s) for this statement missing.**

Answer: Suggestion attended. The reference will be added.

**P. 3 lines 66-67: suggest to reference the study of Ramacher et al. (2019) on local scale for Baltic Sea ports.**

Answer: Suggestion attended. The reference will be added.

**P. 5 line 150: "ash" - what is this chemically? Please define.**

Answer: Suggestion attended. "Ash" will be defined.

**P. 8 line 232: please replace "lower increases contributions" by "lower positive contributions".**

Answer: Suggestion attended.

Conclusions: the word "verify" is used several times in the conclusions section (P. 10, line 321; P. 10, line 324; P. 11, line 330). Verification implies the comparison of model results to the true values, which are not known. Please change wording.

Answer: Suggestion attended. The word "verify" will be changed by "observe" and "detect".

**P. 11 line 340: what about the code availability of STEAM 3? Please include a statement here.**

Answer: Suggestion attended. The following statement will be added:" STEAM model is intellectual property of the Finnish Meteorological Institute and is not publicly available".

**P. 15 lines 469-471: the citation of Marelle et al. is incomplete.**

Answer: Suggestion attended. The reference will be changed.

**Table 1 and Table 2: "tonne y-1" is not SI unit.**

Answer: Given that the values are high, it is usual to present the results in above referred units. We maintained the units to be more coherent with the literature.

**Figure 1 and Figure 2: please use SI units in labels, axis annotations and captions.**

Answer: Given that the values are high, it is usual to present the results in above referred units. We maintained the units to be more coherent with the literature.

**Figure 4f: what is the cause for high O3 values along the North African coast overwater?**

Answer: Thank you for your comment. There are no O3 sinks in this region.

**AUTHOR'S RESPONSES TO REFEREE #3:**

We thank Referee #3 for the positive evaluation and for taking the time to read our paper and giving us valuable comments to improve the manuscript. Following the reviewer remarks, we addressed the comments and questions in detail below:

**Major comment**

My main criticism of the manuscript is that the results on additional exceedances of air quality standards due to ship emissions (which are potentially of interest to policymakers) depend strongly on the quality of the modelled fields. Only if they give a good representation of the actual air quality and exceedances, then the difference between the results with and without ship emissions can be trusted. I therefore believe that the authors need to include a comparison of the modelled concentrations and exceedances for the scenario including ship emissions to those measured by in-situ air quality net-works to demonstrate that they are close enough to reality to make interpretation of delta exceedances worthwhile.

Answer: Thank you for your comments. To support our results, model output  $PM_{2.5}$ ,  $PM_{10}$  and NO2 concentrations for the S-SCN scenario were compared with data from the monitoring stations of the EU Member States reported by the European Environmental Agency for 2015. Moreover, comparisons between the modelling reference results reported by EMEP for the year 2015 were also compared with the data from the monitoring stations. Annual mean concentrations observed in 139 stations for PM2.5, 337 stations for PM10 and 446 stations for NO2 were compared with the model results in time and space. Information about model validation will be added in the Methods section as follows: "...and recent studies that used the model to assess the effects of shipping emissions (Jonson et al., 2015, 2017; Turner et al., 2017). To support the results of the present study, model output PM2.5, PM10 and NO2 concentrations for the S-SCN scenario were compared with data from the monitoring stations of the EU Member States reported by the European Environmental Agency for 2015 (EEA, 2020). Moreover, comparisons between the modelling reference results reported by EMEP for the year 2015 (Norwegian Meteorological Institute, 2019) were also compared with the data from the monitoring stations. Annual mean concentrations observed in 139 stations for PM2.5, 337 stations for PM10 and 446 stations for NO2 were compared with the model results in time and space. Table 1 summarizes the model quality indicators (Pearson correlation coefficient (Pearson's r), Mean Bias Error (MBE), Mean Absolute Error (MAE) and Root Mean Square Error (RMSE)), for the present study estimations and the reference results reported by EMEP. Similar quality indicators

were obtained for the comparison of the results of the present study and the reference results of EMEP, which indicates that the model simulations were well executed. Although the correlations obtained were moderate positive correlations (Pearson's r > 0.5) for all pollutants, the errors obtained were smaller than those reported in the literature (Monteiro et al., 2018), which make our results acceptable."

| Indicators        |                   | This study       |                 | E۸                | AEP reference    | ce              |
|-------------------|-------------------|------------------|-----------------|-------------------|------------------|-----------------|
|                   | PM 2.5 | PM 10 | NO 2 | PM 2.5 | PM 10 | NO 2 |
| Pearson's r       | 0.57              | 0.55             | 0.70            | 0.64              | 0.55             | 0.67            |
| MBE a  | 1.32              | 19.51            | 5.78            | 0.34              | 18.70            | 5.19            |
| MAE b  | 2.86              | 19.55            | 8.70            | 2.81              | 18.74            | 9.18            |
| RMSE c | 3.62              | 20.83            | 11.24           | 3.59              | 20.11            | 11.90           |

**Table 1.** Model quality indicators for the present study estimations and the reference resultsreported by EMEP.

a Mean Bias Error; b Mean Absolute Error; c Root Mean Square Error

Moreover, a comparison between the exceedances for the modelled scenario including ship emissions and those calculated with the data from the monitoring stations was also made. We were able to compare only the exceedances for  $PM_{2.5}$ ,  $PM_{10}$  and  $NO_2$  since we didn't have daily  $SO_2$  concentrations data from the monitoring stations. Information about the comparison of the exceedances found with the model and with the data from the stations will be added in the Methods section as follows: "...and  $PM_{10}$  (20 µg m-3 for annual mean) (European Comission, 2018; WHO, 2018). To support the results of the present study,  $PM_{2.5}$ ,  $PM_{10}$  and  $NO_2$  exceedances found for the S-SCN scenario were compared with those calculated with data from the monitoring stations of the EU Member States (EEA, 2020). For  $PM_{2.5}$ , the exceedances to the WHO guideline found with the modelled data represented more than 60% of the exceedances calculated with the data from the stations. Regarding  $PM_{10}$ , a small agreement was found, with only 11% of the exceedances found for the modelled data. However, for  $NO_2$  all the exceedances were estimated with the modelled data. According to these results, the model seems to predict with good reliability the exceedances of  $PM_{2.5}$  and  $NO_2$ . For  $PM_{10}$  the results need to be used with caution."

**Minor comments**

While the manuscript is overall well written, it would benefit from proof reading by a native speaker.

Answer: Suggestion attended. The manuscript will be review by a native speaker.

**page 4, line 120: Are Sahara dust emissions and NOx from lightning really taken from the NCAR fire inventory?**

Answer: No. Only the forest fire emissions were taken from the NCAR fire inventory. To improve the comprehension of the sentence we decided to delete the part of "from the Fire INventory from NCAR version 1.5" and keep the reference of the inventory and add the Simpson et al. (2012) reference where more information about these emissions can be found.

**page 7, line 204 and figures: I think it is stated nowhere that when you talk about concentrations, that always means at the surface (I assume)**

Answer: Thank you for your comment. The concentrations are surface concentrations. Modifications will be introduced in the line referred and the figure legend.

**page 8, line 251 and following: I'm a bit confused by this discussion of the origins of the seasonality. It sounds as if it is not really clear what the origin is, but don't you have all the information on the magnitude of emissions from STEAM so that you can give clear answers on what drives the seasonality?**

Answer: Thank you for your comment. This analysis was related to the concentrations. Although we had all the information on the magnitude of emissions from STEAM, there are other factors that can influence the concentrations over the seasons. To understand if statistically significant differences in concentrations between the various seasons exist, the non-parametric test Kruskal-Wallis test was used to compare multiple samples (the four seasons) and the nonparametric Wilcoxon signed-rank test was used to compare related samples (two by two). Moreover, as our previous analyses claim it was during spring and summer that were registered the highest emission amounts. Information about the statistical analyses above referred will be added in the Methods section as follows: "The non-parametric test Kruskal-Wallis for multiple samples (the four seasons) and the non-parametric Wilcoxon signed-rank test for two by two samples analyses performed at the 95% confidence interval level were used to detect statistically significant variations for all pollutants in the seasonal concentration data." Moreover, we will change the Results section as follows: "Regarding the seasonal concentration data, statistically significant variations were found for all pollutants across all seasons (p-values < 0.05). In fact, according to the model results, the higher contributions of shipping emissions to the concentrations levels were registered during spring and summer periods (warm season). This pattern seems to be related to the increase in ship traffic during summer due to better meteorological conditions that allow better navigation conditions, which increases the traffic and subsequently the emissions and atmospheric pollution. Moreover, during summer months,

the number of passenger ships tends to increase (due to recreational travel), especially in the Mediterranean Sea, which led to an increase of shipping emissions and their contributions to the pollutant's concentration levels."

page 10, line 299: The discussion on uncertainties and limitations is very general indeed and mainly lists the obvious. I think that the comparison to real data will make this section also more relevant.

Answer: Thank you for your comment. Comparison with real data will be added according to a previous comment.

Figure 1: I'm not sure that it makes really sense to show all these figures here they all look the same with the colour scale chosen and I do not see what I can learn from 8 figures which I cannot already see in the first.

Answer: Thank you for your comment. In a previous version of the manuscript, we had the same scale for all pollutants which allowed to see the differences in the quantities of each pollutant. We will change it back to that version.

**Shipping emissions in the Iberian Peninsula and its impacts on air quality**

Rafael A.O. Nunes1, Maria C.M. Alvim-Ferraz1, Fernando G. Martins1, Fátima Calderay-Cayetano2, Vanessa Durán-Grados2, Juan Moreno-Gutiérrez2, Jukka-Pekka Jalkanen3, Hanna Hannuniemi3, Sofia I.V. Sousa1

[revised manuscript text omitted]

- $\begin{array}{ll} \text{165} & \text{mean}, \text{SO}_2 \left(125 \ \mu \text{g m}^{-3} \ \text{for daily mean}\right), \text{PM}_{2.5} \left(25 \ \mu \text{g m}^{-3} \ \text{for annual mean}\right) \ \text{and} \ \text{PM}_{10} \ (40 \ \mu \text{g m}^{-3} \ \text{for annual mean}); \ \text{and} \ \text{ii}) \\ \text{WHO air quality guidelines for NO}_2 \ (40 \ \mu \text{g m}^{-3} \ \text{for annual mean}), \ \text{SO}_2 \left(20 \ \mu \text{g m}^{-3} \ \text{for daily mean}\right), \ \text{O}_3 \ (\text{SOMO35 yearly sum}); \ \text{and} \ \text{ii}) \\ \end{array}$

5

**Deleted:** from the "Fire INventory from NCAR version 1.5" **Deleted:** (Wiedinmyer et al., 2011) of the daily maximum of 8 h running average over 35 ppb in ppb per days), PM2.5 (10 µg m-3 for annual mean) and PM10 (20
µg m-3 for annual mean) (European Comission, 2018; WHO, 2018). To support the results of the present study, PM2.5, PM10 and NO2 exceedances found for the S-SCN scenario were compared with those calculated with data from the monitoring stations of the EU Member States (EEA, 2020). For PM2.5, the exceedances to the WHO guideline found with the modelled data represented more than 60% of the exceedances found for the modelled data. However, for NO2 all the exceedances found for the exceedances found for the modelled data. However, for NO2 all the exceedances were estimated with the modelled data. According to these results, the model seems to predict with good reliability the exceedances of PM2.5 and NO2. For PM10 the results need to be used with caution.

**3 Results and Discussion**

**3.1 Shipping emissions - spatial and seasonal variation**

Table 2 summarizes the amount of emitted air pollutants from shipping and from land-based anthropogenic sources. 180 Comparing NOx and SOx, shipping emissions with land-based emissions, on average the first were lower than the latter. Despite this, if NOx and SOx shipping emissions were added to the land-based emissions, the total would increase by 45% and 62%, respectively. Moreover, compared with emissions from the SNAP of road transport (660 ktonnes y-1 of NOx and 7.1 ktonnes y-1 of SOx), the emitted amounts of NOx and SOx from shipping were 1.1 and 51.3 times higher, respectively. These results show the importance of shipping emissions for these two pollutants.

- 185 Fig: 1 shows the annual mean shipping emissions of CO, CO2, SOx, NOx, EC, OC, sulphates and ash (a component of the PM emitted by ships and depends on the content of marine fuels) for the Iberian Peninsula in 2015 in a 0.1°x0.1° grid cells (approximately 10 x 10 km2). Details about emission factors used in STEAM can be found in Jalkanen et al. (2009), Jalkanen et al. (2012) and Jonson et al. (2015). As can be seen, the spatial distribution was similar for all pollutants. In general, the highest emissions were established along the west coast of the Iberian Peninsula (including all Portuguese coast), in the Strait
- 190 of Gibraltar and in the Mediterranean Sea, especially close to the African coast, which is consistent with world shipping traffic density (Fig: A1). It is important to emphasise that the grid cells along the coast where ports are located had also higher emissions due to hotelling activities. Although emissions during hotelling only represent a slight part of the total shipping emissions, port areas are significant receptors of these emissions due to the concentration of ships for long periods of time in some cases (Nunes et al., 2017a). The annual average and highest intensities for NOx and SOx reported from researches in
- 195 Asian Region are present in Table 3 (Chen et al., 2016a, 2017; Fan et al., 2016). In general, the average intensities that were reported for the Asia were considerably higher than those found in this study. It was possible to identify in the present study two main hubs given the high emissions intensity: Valencia Port and the Strait of Gibraltar. At Valencia Port, ash, CO, EC and OC had the highest values, respectively, 1.46E-01 tonnes/yr/km2, 1.85 tonnes/yr/km2, 1.99E-01 tonnes/yr/km2 and 5.09E-01 tonnes/yr/km2. At the Strait of Gibraltar, CO2, NO3, sulphate and SO3 had the highest values, respectively, 1330 tonnes/yr/km2,

[revised manuscript text omitted]

**630 **Table 1.** Model quality indicators for the present study estimations and for the reference results reported by EMEP.**

a Mean Bias Error; b Mean Absolute Error; c Root Mean Square Error

**Table 2.** Annual mean amounts of emitted air pollutants from shipping and from land-based anthropogenic sources during 2015 (in tonne  $y^{-1}$ )

I

| Pollutant       | Shipping | Land-based emissions a | Road transport emissions |
|-----------------|----------|-----------------------------------|--------------------------|
| Ash             | 3.3E+03  | -                                 | -                        |
| EC              | 4.5E+03  | -                                 | -                        |
| OC              | 1.1E+04  | -                                 | -                        |
| NOx             | 7.1E+05  | 1.6E+06                           | 6.6E+05                  |
| SO x | 3.6E+05  | 5.8E+05                           | 7.1E+03                  |
| Sulphate        | 3.2E+04  | -                                 | -                        |
| CO 2 | 3.0E+07  | -                                 | -                        |
| со              | 4.9E+04  | 3.6E+06                           | 5.7E+05                  |
| Total           | 3.1E+07  | -                                 | -                        |

635 a Emissions from 11 SNAP sectors, namely, public electricity and heat production, industry, other stationary combustion sources, fugitive emissions, solvents, road transport, aviation, off-road sources, waste, agriculture livestock, agriculture other sources and other sources.

|                            | Port/sea area       | NOx     |                                | SOx     |                                |
|----------------------------|---------------------|----------------|--------------------------------|----------------|--------------------------------|
| Study               |                     | Annual average | Highest
value | Annual average | Highest
value |
| Chen et al. (2016a) | Tianjin Port | 5.06    | 1.51E+03                       | 7.14           | 1.79E+03                |
| Chen et al. (2017)         | Qingdao Port        | 1.83    | =                              | 1.42    | =                              |
| Fan et al. (2016)          | East China Sea      | 1.0     | 1.0E+04                 | 1.90    | 1.30E+03                       |
| Ng et al. (2013)    | Hong Kong           | =              | 1.1E+02                 |                | 2.0E+02                 |

Table 3. Annual average and highest intensities of NO3 and SO3 (in tonnes/yr/km2) reported from researches in Asian Region.

| Pollutant       | Spring  | Summer  | Autumn  | Winter  | Total   |
|-----------------|---------|---------|---------|---------|---------|
| Ash             | 0.85    | 0.87    | 0.83    | 0.77    | 3.3     |
| EC              | 1.2     | 1.2     | 1.1     | 1.0     | 4.5     |
| OC              | 2.9     | 3.0     | 2.8     | 2.6     | 11      |
| NOx             | 1.8E+02 | 1.9E+02 | 1.8E+02 | 1.6E+02 | 7.1E+02 |
| SOx             | 92      | 94      | 91      | 85      | 36      |
| Sulphate        | 8.3     | 8.4     | 8.1     | 7.6     | 32      |
| CO 2 | 7.8E+03 | 8.0E+03 | 7.6E+03 | 7.0E+03 | 3.0E+04 |
| со              | 13      | 13      | 12      | 12      | 49      |
| Total           | 8.3E+03 | 8.3E+03 | 7.9E+03 | 7.2E+03 | 3.1E+04 |

 $\label{eq:table_formula} \textbf{Table \underline{4}}. Seasonal amounts of emitted air pollutants from shipping in the Iberian Peninsula in 2015 (in tonne y^1)$

---

## Author Response (AR2)

Dear Editor,

Please fin attached the point-by-point response to the reviews.
Thank you.

Kind regards,

Sofia Sousa

**AUTHOR'S RESPONSES TO REFEREE #1:**

**Addition of comparisons to station data is valuable, and including comparison of exceedances helps to put results into perspective. However, from reading the text, it is not clear to me how the results were calculated:**
**"For PM2.5, the exceedances to the WHO guideline found with the modelled data represented more than 60% of the exceedances calculated with the data from the stations".**
**What does this mean? Is the number of exceedances compared or the day of exceedance, and how are the different cases of finding / not finding exceedances in model / measurement treated? I could imagine cases where the model finds an exceedance but the measurements do not and the other way round and would not know how to combine these results into one percentage. Please provide more information here.**
Answer: Suggestion attended. The annual exceedances of PM2.5, PM10 and NO2 found simultaneously with the modelled S-SCN scenario and with data from the monitoring stations of the EU Member States were compared. Clarifications regarding these calculations were introduced in the text. Please see lines 165-167.

**Please also add units to Table 1 where appropriate..."**

Answer: Suggestion attended. Please see Table 1.

**AUTHOR'S RESPONSES TO REFEREE #2:**

**The manuscript was considerably improved compared to the earlier version. Two remaining issues should be addressed before I can recommend publication in Atmospheric Chemistry and Physics:**

**1) The authors provided a detailed elaboration on the sources of uncertainties in the STEAM 3 inventory in their final response. The authors are asked to include the details on uncertainties in STEAM also in section 3.3 "Uncertainties and Limitations", so that other researchers can dedicate efforts to quantify, how the specified uncertainties of ship emissions affect modelled air pollutant concentrations.**

Answer: Suggestion attended. Please see lines 344-353.

**2) VOC emissions from ships were not included in the study. The chemical regime in the atmosphere along the ship tracks in the Mediterranean is known to be VOC sensitive (Beekmann and Vautard, 2010), implying that ozone production is very sensitive to emission of reactive VOC from the ships travelling there. This is an additional limitation of the present study, specifically when quantifying the SOMO35 indicator. This must be clearly stated in section 3.3 "Uncertainties and Limitations".**
**References:**
**Beekmann, M. and Vautard, R.: A modelling study of photochemical regimes over Europe: robustness and variability, Atmos. Chem. Phys., 10, 10067-10084, https://doi.org/10.5194/acp-10-10067-2010,**
**2010.**

Answer: Suggestion attended. Please see lines 353-358.